# Test Where Decisions Matter:
# Importance-driven Testing for Deep Reinforcement Learning

**Stefan Pranger,**[1] **Hana Chockler,**[2] **Martin Tappler,**[3] **Bettina Könighofer**[1]
[1]Institute of Information Security,
Graz University of Technology
[2]Kings College London
[3]Institute of Computer Engineering,
TU Wien
{stefan.pranger,bettina.koenighofer}@tugraz.at
hana.chockler@kcl.ac.uk
martin.tappler@tuwien.ac.at

## Abstract

In many Deep Reinforcement Learning (RL) problems, decisions in a trained policy vary in significance for the expected safety and performance of the policy. Since RL policies are very complex, testing efforts should concentrate on states in which the agent's decisions have the highest impact on the expected outcome. In this paper, we propose a novel model-based method to rigorously compute a ranking of state importance across the entire state space. We then focus our testing efforts on the highest-ranked states. In this paper, we focus on testing for safety. However, the proposed methods can be easily adapted to test for performance. In each iteration, our testing framework computes optimistic and pessimistic safety estimates. These estimates provide lower and upper bounds on the expected outcomes of the policy execution across all modeled states in the state space. Our approach divides the state space into safe and unsafe regions upon convergence, providing clear insights into the policy's weaknesses. Two important properties characterize our approach. (1) Optimal Test-Case Selection: At any time in the testing process, our approach evaluates the policy in the states that are most critical for safety. (2) Guaranteed Safety: Our approach can provide formal verification guarantees over the entire state space by sampling only a fraction of the policy. Any safety properties assured by the pessimistic estimate are formally proven to hold for the policy. We provide a detailed evaluation of our framework on several examples, showing that our method discovers unsafe policy behavior with low testing effort.

## 1 Introduction

Deep reinforcement learning (RL) [1] is a powerful method for training policies that complete tasks in complex environments. Due to the high potential of RL in safety-critical domains, such as autonomous driving [2], ensuring the reliability of its safety-critical properties is becoming increasingly vital. Formal verification provides provable correctness guarantees [3]. However, the most significant challenge in the formal verification of RL policies is scalability, which limits its current applicability [4]. As for conventional software, a complete safety evaluation by exhaustively testing a policy's decisions is infeasible. Hence, it is necessary to establish as much confidence as possible in a policy with a limited testing budget.

38th Conference on Neural Information Processing Systems (NeurIPS 2024).

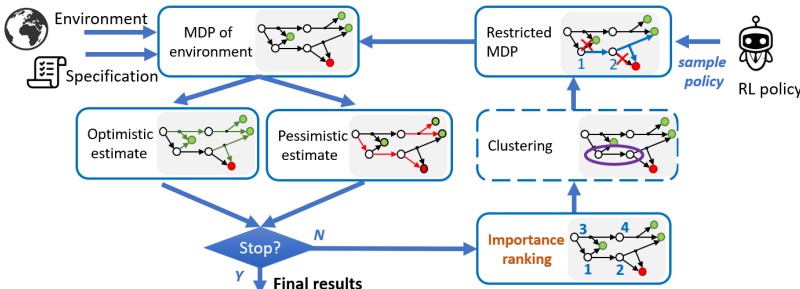

Figure 1: High-level view of the algorithm for importance-driven testing for RL.

We propose a novel model-based testing framework for RL policies, which tests policies in the states where *their decisions matter most*. We follow the insights from Chockler et al. [5] that not all decisions hold equal significance on the expected safety and performance of a policy. Decisions in certain states may have a significant impact on the overall expected outcome of the policy, while in other states, the impact may not be as severe or critical. The core of our algorithm is a *ranking of the importance of states of the environment*. This ranking is based on the difference that the decision in a particular state makes on the expected overall performance (e.g., accumulated reward) or safety of the policy. For lack of space, we focus on safety from here on. The proposed method can easily be adapted to evaluate the agent's performance, which we discuss in Appendix D.

Figure 1 outlines our algorithm. The inputs to our algorithm are a model of the environment in the form of a Markov decision process [6], a formal safety specification $\varphi$, and an RL agent in the form of a deterministic policy. In each iteration, our algorithm computes *optimistic and pessimistic estimates*, which provide lower and upper bounds for the expected probability of satisfying the safety specification over all possible policies. The algorithm terminates if the maximal difference between the estimates gets below some threshold or a maximal number of executed test cases is reached. As long as the stopping criterion is not met, the algorithm computes an *importance ranking* of the states. The higher the rank of a state, the more influence the decision in that state has for satisfying or violating the safety specification. Next, the most important decisions of the policy are sampled and used to fix the decisions, thus *restricting* the MDP. The algorithm continues with the restricted MDP to iteratively refine the estimates. Our testing framework can be modified through an optional step by *clustering the highly ranked states*. A fraction of test cases is then uniformly selected from the individual clusters. The intuition behind clustering is that the agent is likely to behave similarly in comparable situations. Following this intuition, we mark all states in a cluster as safe if all tested states of this cluster are verified to be safe. Otherwise, the entire cluster is marked as unsafe. This increases the scalability of our testing approach since only a fraction of each cluster needs to be tested for deriving test verdicts for all states in the cluster. However, since not all decisions in a cluster are sampled, unsafe policy behavior can be missed.

Our algorithm provides the following *benefits*:

- Optimal Test-Case Selection: At any time in the testing process, our approach evaluates the policy in the states that are most critical for safety.

- Guaranteed Safety: A pessimistic estimate provides a formal verification guarantee: under the given model it is guaranteed that if the pessimistic estimate for a given state satisfies the testing criteria, then the agent's policy is formally verified from that state.

- Highlighting the most important decisions is a central technique in explainable AI [7, 8]. We provide a rigorous method to compute an importance ranking. Simpler policies that only use the top-ranked decisions can help understand the policy's decision-making [5].

- The iterative nature of our approach can be used in a debugging process to construct a safety frontier: if a sampled decision in a certain state is evaluated to be unsafe, the next ranking iteration assigns higher importance to the predecessor states to be tested next. Thus, the unsafe region around a safety hazard grows until it reveals all states where the policy behaves unsafely.

- Upon convergence, our approach partitions the state space into safe and unsafe regions. The identified unsafe regions offer interpretable insights into the policy's current weaknesses.

## 1.1 Related Work

**Evaluation of RL policies.** Off-policy evaluation (OPE) [9, 10, 11] aims to estimate the expected performance of a newly trained policy by using executions of a previously trained policy. In contrast, our approach estimates the performance of the policy under test by computing the best-possible (optimistic) estimate and the worst-possible (pessimistic) estimate in the current MDP. In contrast to OPE, our framework provides formal verification guarantees over the entire state space: any safety property assured by the pessimistic estimate is formally proven to hold. Several recent works proposed evaluation strategies to analyze RL policies [12], by adapting software testing techniques to RL. Various approaches apply fuzz or search-based testing as a basis to find safety-critical situations in the black-box environment [13, 14, 15, 16], in which to test the policy. Most efforts of the testing community focused on selecting test cases that falsify safety with high probability [15, 17]. These methods effectively reveal unsafe behavior, but they do not provide safety assurance from non-failing tests, as they lack proper notions of coverage. In contrast, our testing approach is model-based. Model-based testing of probabilistic systems was proposed in [18]. To the best of our knowledge, there is no model-based testing approach for RL policies with formalized criteria of completeness. That is, we are the first to propose safety estimates with formal interpretations which form the basis of our test-case generation.

**Model-based formal methods for model-free RL.** Several recent works have proposed approaches for developing RL controllers by combining model-based formal methods and model-free RL. The appeal of this combination lies in the strengths of each approach: model-based methods offer formal safety and correctness guarantees, while model-free RL demonstrates superior scalability and yields high-performance controllers by learning from the full-order system dynamics [19, 20]. Most of the existing work in this area addresses the problem of safe exploration in RL [21, 22]. To the best of our knowledge, our work is the first to employ similar techniques for analyzing a trained policy.

**Importance ranking.** Ranking policy decisions has been proposed for explaining and simplifying RL policies. In [5], the ranking is based on statistical fault localization computed on a set of executions of the original policy and its small perturbations. A continuation of this work [23] uses an average treatment effect to rank policy decisions. In contrast, we provide a rigorous method to compute the importance ranking. Our estimates consider any possible behavior of the policy over the entire state space. Thus, the estimates provide strong verification guarantees. Similarly to ranking policy decisions, [24] rank the importance of individual neurons in a network to assess coverage of a given test set.

## 2 Background

**Markov Decision Process.** A *Markov decision process* (MDP) [25] $\mathcal{M} = (\mathcal{S}, \mathcal{A}, \mathcal{P}, \mu)$ is a tuple with a finite state set $\mathcal{S}$, a finite set $\mathcal{A} = \{a_1 \ldots, a_n\}$ of actions, a probabilistic transition function $\mathcal{P} : \mathcal{S} \times \mathcal{A} \times \mathcal{S} \to [0, 1]$, and a probability distribution of initial states $\mu : \mathcal{S} \to [0, 1]$. An *execution* (or path) is a finite or infinite sequence $\rho = s_0, a_0, s_1, a_1 \ldots$ with $\mathcal{P}(s_i, a_i, s_{i+1}) > 0$ and $\mu(s_0) > 0$. A (memoryless deterministic) *policy* $\pi : \mathcal{S} \to \mathcal{A}$ is a function mapping states to actions. $\Pi$ denotes the set of all memoryless deterministic policies. Applying $\pi$ to an MDP $\mathcal{M}$ induces a Markov chain (MC) $\mathcal{M}^\pi$. An execution in $\mathcal{M}^\pi$ is a sequence $\rho = s_0, s_1, s_2, \ldots$ with $\mathcal{P}(s_i, \pi(s_i), s_{i+1}) > 0$. $\mathbb{P}_s^\pi$ denotes the unique probability measure of $\mathcal{M}^\pi$ over infinite executions starting in $s$.

**Probabilistic Model Checking.** Probabilistic model checking [3] computes the probabilities of satisfying a temporal-logic formula $\varphi$ over a finite or infinite horizon. We define the properties below with a bound $n$. For the unbounded horizon, $n = \infty$. For a given MDP $\mathcal{M}$, a policy $\pi$, and a property $\varphi$ in Computation Tree Logic (CTL) [3], model checking computes the following probabilities:

- $\mathbb{P}_{\mathcal{M}^\pi, \varphi} : \mathcal{S} \times \mathbb{N} \to [0, 1]$ is the expected probability to satisfy $\varphi$ state $s \in \mathcal{S}$ within $n$ steps in $\mathcal{M}^\pi$.
- $\mathbb{P}_{\mathcal{M}, \varphi}^{\mathsf{max}}(s, n) = \max_{\pi \in \Pi} \mathbb{P}_{\mathcal{M}^\pi, \varphi}(s, n)$ is the *maximal* expected probability *over all policies in* $\Pi$ from a state $s$ within $n$ steps.
- $\mathbb{P}_{\mathcal{M}, \varphi}^{\mathsf{min}}(s, n) = \min_{\pi \in \Pi} \mathbb{P}_{\mathcal{M}^\pi, \varphi}(s, n)$ is the *minimal* expected probability *over all policies in* $\Pi$ from a state $s$ within $n$ steps.

For the remainder of this paper, let $\varphi$ be a formula in the safety fragment of CTL. Using $\varphi$ and a user-defined safety threshold $\delta_\varphi$, we define safety objectives as follows:

**Algorithm 1** Importance–driven model-based testing (IMT)
***
**Input**: MDP $\mathcal{M}$, policy $\pi$, safety objective $\langle \varphi, \delta_\varphi \rangle$
**Parameters**: # samples $m$, safety threshold $\delta_\varphi$, minimal difference $\varepsilon_\varphi$
**Output**: failure states $\mathcal{S}_f \subseteq \mathcal{S}$, safe states $\mathcal{S}_s \subseteq \mathcal{S}$, estimates $e_{opt} : \mathcal{S} \to \mathbb{R}$ and $e_{pes} : \mathcal{S} \to \mathbb{R}$

1: $\mathcal{M}^{(0)} \leftarrow \mathcal{M}; \mathcal{S}_u \leftarrow \mathcal{S}; \mathcal{S}_f \leftarrow \emptyset; \mathcal{S}_s \leftarrow \emptyset; i \leftarrow 0$
2: **loop**
3:     $e_{opt}, e_{pes} \leftarrow \text{computeEstimates}(\mathcal{M}^{(i)})$
4:     $\mathcal{S}_s \leftarrow \mathcal{S}_s \cup \{s \in \mathcal{S}_u \mid e_{pes}(s) \geq \delta_\varphi\}$
5:     $\mathcal{S}_f \leftarrow \mathcal{S}_f \cup \{s \in \mathcal{S}_u \mid e_{opt}(s) < \delta_\varphi\}$
6:     $\mathcal{S}_u \leftarrow \mathcal{S}_u \setminus \{s \in \mathcal{S}_u \mid e_{opt}(s) < \delta_\varphi \vee e_{pes}(s) \geq \delta_\varphi\}$
7:     **if** $[\max_s(e_{opt}(s) - e_{pes}(s)) < \varepsilon_\varphi]$ **then**
8:         **stop**
9:     **end if**
10:    $\mathcal{S}_{rank} \leftarrow [\text{computeRanking}(\mathcal{M}^{(i)}, m)]$
11:    $\{(s_1, a_1) \ldots (s_m, a_m)\} \leftarrow \text{samplePolicy}(\pi, \mathcal{S}_{rank})$
12:    $\mathcal{M}^{(i+1)} \leftarrow \text{restrictMDP}(\mathcal{M}^{(i)}, \{(s_1, a_1) \ldots (s_m, a_m)\})$
13:    $i \leftarrow i + 1$
14: **end loop**
15: **return** $\mathcal{S}_f, \mathcal{S}_s, e_{opt}, e_{pes}$
***

**Definition 2.1** (Safety objective). Given an MDP $\mathcal{M} = (\mathcal{S}, \mathcal{A}, \mathcal{P}, \mu)$, a safety property $\varphi$, and a threshold $\delta_\varphi \in [0, 1]$. A *safety objective* is a tuple $\langle \varphi, \delta_\varphi \rangle$. A policy $\pi$ satisfies $\langle \varphi, \delta_\varphi \rangle$ from a given state $s \in \mathcal{S}$ within $n$ steps if $\mathbb{P}_{\mathcal{M}^\pi, \varphi}(s, n) \geq \delta_\varphi$.

**Reinforcement Learning.** An RL [1] agent learns a task via interactions with an unknown environment modeled by an MDP $\mathcal{M}$ with an associated reward function $\mathcal{R} : \mathcal{S} \to \mathbb{R}$. In each state $s \in \mathcal{S}$, the agent chooses an action $a \in \mathcal{A}$, the environment then moves to a state $s'$ with probability $\mathcal{P}(s, a, s')$. The return $\texttt{ret}_\rho$ of an execution $\rho$ is the discounted cumulative reward defined by $\texttt{ret}_\rho = \Sigma_{t=0}^\infty \gamma^t \mathcal{R}(s_t)$, using the discount factor $\gamma \in [0, 1]$. The objective of the agent is to learn a deterministic memoryless *optimal policy* $\pi^\star$ that maximizes the expectation of the return.

## 3 Importance-driven Testing for RL

In this section, we will describe our framework for importance-driven model-based testing, which we abbreviate with IMT. An overview of our algorithm is depicted in Fig. 1. Its central elements are the computation of the estimates and the importance ranking that guides the selection of the test cases. In Sec. 3.1 we discuss IMT in detail, and in Sec.3.2 we discuss its extension with clustering.

### 3.1 Importance-driven Model-Based Testing

Alg. 1 gives the pseudo-code of our approach for importance-driven safety testing. Our algorithm evaluates a policy $\pi$ with respect to a safety objective $\langle \varphi, \delta_\varphi \rangle$ over a horizon of $n$ steps (for the unbounded horizon, $n = \infty$). The algorithm takes as input an MDP $\mathcal{M} = (\mathcal{S}, \mathcal{A}, \mathcal{P}, \mu)$, a policy under test $\pi : \mathcal{S} \to \mathcal{A}$, and a safety objective $\langle \varphi, \delta_\varphi \rangle$. It returns as result a classification of states into safe and failure states ($\mathcal{S}_s$ and $\mathcal{S}_f$, respectively), and the optimistic and pessimistic estimates for all states in the state space ($e_{opt} : \mathcal{S} \to [0, 1]$ and $e_{pes} : \mathcal{S} \to [0, 1]$, respectively), which are derived as the expected maximal and minimal probability of satisfying the safety objective $\langle \varphi, \delta_\varphi \rangle$.

**Safety estimates.** In Line 3, IMT computes the safety estimates for the current (restricted) MDP $\mathcal{M}^{(i)}$. The optimistic estimate $e_{opt}(s, n)$ is the maximal expected probability of satisfying $\varphi$ for an execution in $\mathcal{M}^{(i)}$ from a given state $s$ within a $n$ steps quantified over all policies. Similarly, the pessimistic estimate $e_{pes}(s, n)$ is the minimal expected probability of satisfying $\varphi$.This yields the following definition:

**Definition 3.1** (Safety estimates). For a given MDP $\mathcal{M} = (\mathcal{S}, \mathcal{A}, \mathcal{P}, \mu)$, a given safety property $\varphi$, and a given number of $n$ steps, the *optimistic* and *pessimistic safety estimate* $e_{opt}, e_{pes} : \mathcal{S} \times \mathbb{N} \to [0, 1]$

are defined as follows:
$$\forall s \in \mathcal{S}\colon e_{opt}(s,n) = \mathbb{P}^{\mathsf{max}}_{\mathcal{M},\varphi}(s,n), \text{ and } \quad \forall s \in \mathcal{S}\colon e_{pes}(s,n) = \mathbb{P}^{\mathsf{min}}_{\mathcal{M},\varphi}(s,n).$$

For a state action pair $(s,a)$ and a bound $n$, the maximal expected probability of satisfying $\varphi$ from a state $s$ after executing $a$ is
$$\forall s \in \mathcal{S}, \forall a \in \mathcal{A}\colon e_{opt}(s,a,n) = \sum_{s' \in \mathcal{S}}(\mathcal{P}(s,a,s') \cdot e_{opt}(s',n-1)).$$

Based on the estimates, the algorithm classifies undetermined states from $\mathcal{S}_u$ as verified safe and adds them to $\mathcal{S}_s$ or classifies them as unsafe and adds to the set of failure states $\mathcal{S}_f$ (Lines 4 and 5). A state $s \in \mathcal{S}$ satisfies the safety objective $\langle \varphi, \delta_\varphi \rangle$ if $e_{pes}(s,n) \geq \delta_\varphi$. Note that the pessimistic safety estimate is achieved in an execution that chooses the most unsafe actions in each non-restricted state. Thus, if for a given state $e_{pes}(s,n) \geq \delta_\varphi$, then $\mathbb{P}_{\mathcal{M}^\pi,\varphi}(s,n) \geq \delta_\varphi$ holds. *This highlights the strength of our algorithm: by assuming the worst policy behavior in unrestricted states, we provide verification results without sampling the policy in every state.* A state $s \in \mathcal{S}$ is unsafe if $e_{opt}(s,n) \leq \delta_\varphi$. The optimistic safety probability is achieved in an execution that chooses the safest action in each non-restricted state. Thus, if $e_{opt}(s,n) \leq \delta_\varphi$, the policy $\pi$ cannot pick actions that would yield higher probabilities of satisfying $\varphi$ from $s$.

**Stopping criteria.** In Line 7, the stopping criterion is defined via a user-defined threshold $\varepsilon_\varphi$ for the minimal difference between the estimates. IMT stops if the difference between the optimistic and the pessimistic safety estimate is below the threshold $\varepsilon_\varphi$ for all states, i.e., $\max_s[e_{opt}(s,n) - e_{pes}(s,n)] < \varepsilon_\varphi$. For small values of $e_{pes}$, further restricting the MDP would only marginally change the testing results. Otherwise, IMT continues with sampling the policy and restricting the MDP, as the optimistic and pessimistic estimates are sufficiently different. As an alternative stopping criterion, a user could also define a total testing budget.

**Importance ranking.** In each iteration, IMT computes an importance ranking over all states in the current MDP $\mathcal{M}^{(i)}$ (Line 10). In the following steps, the $m$ most important decisions of the policy are sampled and used to restrict $\mathcal{M}^{(i)}$, which results in refined estimates. The rank of a state $s$ reflects the *maximal difference that a decision can have* on satisfying the safety objective.

**Definition 3.2.** (Importance ranking for safety.) Given an MDP $\mathcal{M} = (\mathcal{S}, \mathcal{A}, \mathcal{P}, \mu)$, a safety property $\varphi$, and a bound $n$, the importance ranking $rank\colon \mathcal{S} \times \mathbb{N} \to \mathbb{R}$ is given as the *maximal difference between the optimistic estimates* with respect to the available actions:
$$\forall s \in \mathcal{S}\colon rank(s,n) = \max_{a,a' \in \mathcal{A}}(e_{opt}(s,a,n) - e_{opt}(s,a',n)).$$

For the importance ranking, we consider the impact of decisions on the *optimistic estimates*. That is, a state $s$ is important if, for some actions $a$ and $a'$, it holds that the expected maximal safety probability that can still be achieved after executing $a$ from $s$ is considerably larger than the probability that can be obtained after executing $a'$ from $s$. The importance ranking returns the set of states $\mathcal{S}_{rank}$ of the $m$ highest ranked states of $\mathcal{M}^{(i)}$.

**Sampling the policy.** In Line 11, IMT samples the decisions of the policy in the highest ranked states $s \in \mathcal{S}_{rank}$ of $\mathcal{M}^{(i)}$. This results in the set $\Gamma = \{(s_1,a_1),\ldots,(s_m,a_m)\}$ with $a_i = \pi(s_i)$.

**Restricting the MDP.** In Line 12, our algorithm restricts $\mathcal{M}^{(i)}$ according to the sampled policy's decisions, i.e., actions not chosen by $\pi$ in the sampled states are removed from $\mathcal{M}^{(i)}$. Given the current MDP $\mathcal{M}^{(i)} = (\mathcal{S}, \mathcal{A}, \mathcal{P}^{(i)}, \mu)$ and the sampled state-action pairs $\Gamma$, the restricted MDP $\mathcal{M}^{(i+1)} = (\mathcal{S}, \mathcal{A}, \mathcal{P}^{(i+1)}, \mu)$ has the following probabilistic transition function:
$$\forall s, s' \in \mathcal{S} \, \forall a \in \mathcal{A}\colon \mathcal{P}^{(i+1)}(s,a,s') = \begin{cases} \mathcal{P}^{(i)}(s,a,s') & s \notin \mathcal{S}_{rank} \text{ or } (s,a) \in \Gamma \\ 0 & \text{else.} \end{cases}$$

In every iteration of the algorithm, more actions in the MDP model become fixed to the actions chosen by $\pi$, leading to more accurate safety estimates for $\pi$, i.e., $e_{pes}(s,n)$ monotonically increases and $e_{opt}(s,n)$ monotonically decreases, for all $s \in \mathcal{S}$.

**Theorem 1.** The algorithm IMT as described in Alg. 1 terminates.

*Proof Sketch.* For a fully restricted MDP, for any $s \in \mathcal{S}$, for any $n$, it holds that $e_{opt}(s,n) = e_{pes}(s,n)$. This holds because a fully restricted MDP is a Markov chain that describes the policy completely. Hence the estimates are the same.

---

**Algorithm 2** IMT with Clustering

---

**Input**: $\mathcal{M}, \pi, \langle \varphi, \delta_\varphi \rangle$
**Parameters**: importance threshold $\delta_i$, testing fraction $\kappa$, testing horizon $n$, $\delta_\varphi$, $\varepsilon_\varphi$
**Output**: $\mathcal{S}_f \subseteq \mathcal{S}, \mathcal{S}_s \subseteq \mathcal{S}, e_{opt} : \mathcal{S} \to \mathbb{R}, e_{pes} : \mathcal{S} \to \mathbb{R}$

  1: // Line 1 — 8 are as in Alg. 1
  9: $\mathcal{S}_{rank} \leftarrow [\text{computeRanking}(\mathcal{M}^{(i)})]$
10: $\mathcal{C} \leftarrow [\text{clusterStates}(\mathcal{S}_{rank}, \delta_i)]$
11: $\{(c_1, v_1) \ldots (c_{|\mathcal{C}|}, v_{|\mathcal{C}|})\} \leftarrow \text{executeTests}(\pi, \mathcal{C}, \kappa)$
12: $\mathcal{M}^{(i+1)} \leftarrow \text{restrictMDP}(\mathcal{M}^{(i)}, \{(c_1, v_1) \ldots (c_{|\mathcal{C}|}, v_{|\mathcal{C}|})\})$
13: // Line 12 — 14 are as in Alg. 1

---

## 3.2 Importance-driven Model-Based Testing with Clustering

In this section, we extend IMT by introducing clustering in Alg. 1. Fig. 1 shows the high-level view of IMT including clustering. For problems with very large state spaces, sampling the agents in all highly-ranked states becomes too expensive. To tackle this scalability issue, we propose to cluster similar states and test only a fixed fraction of the states in each cluster. By doing so, we balance the trade-off between accuracy and scalability: The fewer states from a cluster are tested, the higher the scalability of our testing approach. However, the likelihood that some unsafe behavior of the agent remains undetected increases. Clustering offers the additional advantage that similar states are grouped. Under the assumption that the agent implements a policy that selects the same action in similar situations, IMT most likely detects unsafe behavior by sampling a large enough fraction of each cluster. Alg. 2 states the changes in the pseudo-code for IMT with clustering.

**Clustering.** In Line 10, after computing the importance ranking, IMT performs clustering on all states with an importance ranking value greater than some bound $\delta_i = rank(s.n)$. States are clustered according to their state information and their importance value, i.e., we compute a clustering assignment $cl : \mathcal{S} \times [\delta_i, 1] \to \mathbb{N}$. This gives a partitioning of $\mathcal{S}_{rank}$ into sets of states sharing the same cluster label. Note that any off-the-shelf clustering algorithm can be used to compute the clusters of states. [1]

**Executing tests.** In Line 11, the behavior of the policy in the clustered states is evaluated. From each cluster, a fixed percentage $\kappa$ of states is randomly selected to be tested. To test a state $s$, the agent is executed from $s$ for a certain number of steps. If the safety objective is violated during the execution, $s$ is marked as unsafe and added to $\mathcal{S}_f$. Based on the testing results of the individual states we assign verdicts $v_j$ to the clusters proposing a conservative evaluation of safety. Each cluster $c_j$ with a tested state $s \in \mathcal{S}_f$ is assigned a failing verdict $v_j = \texttt{FAIL}$. Consequently, all states $s \in c_j$ are marked unsafe and added to $\mathcal{S}_f$. Conversely, if a cluster $c_j$ does not contain a single tested state from $\mathcal{S}_f$, it is assigned a safe verdict $v_j = \texttt{SAFE}$, and its states are added to $\mathcal{S}_s$.

**Restricting the MDP.** In Line 12, IMT restricts $\mathcal{M}^{(i)}$ in all states that belong to a cluster $c_j$ by turning the states into sink states. Additionally, if a cluster $c_j$ has the verdict $v_j = \texttt{FAIL}$, all states are considered a safety violation.

**The effects of clustering.** Since IMT with clustering only tests a fraction $\kappa$ of each individual cluster, the size and quality of the computed clusters affect the testing process. Clusters that are too large can lead to unnecessary testing efforts, as safe behavior might be deemed unsafe due to conservative evaluation. Additionally, if a cluster contains states that are not sufficiently similar, IMT with clustering may fail to detect unsafe behavior in the policy.

**Complexity Analysis.** We discuss the computational complexity of a single iteration of IMT. The safety estimates are computed via value iteration in $\mathcal{O}(poly(size(\mathcal{M})) \cdot n)$, with $n$ being the bound for the objective [3]. The computation of the ranking only requires sorting of the computed estimates and thus requires $\mathcal{O}(|\mathcal{S}| \log |\mathcal{S}|)$ time. The subsequent restriction of $\mathcal{M}$ is linear in the number of actions present in $\mathcal{M}$, i.e. $\mathcal{O}(|\mathcal{S}| \cdot |\mathcal{A}|)$. Lastly, the complexity of sampling the policy is dependent on the network architecture and the costs for clustering the state space depend on the chosen algorithm.

---

[1]For vision-based state spaces, deep clustering approaches could be used [26].

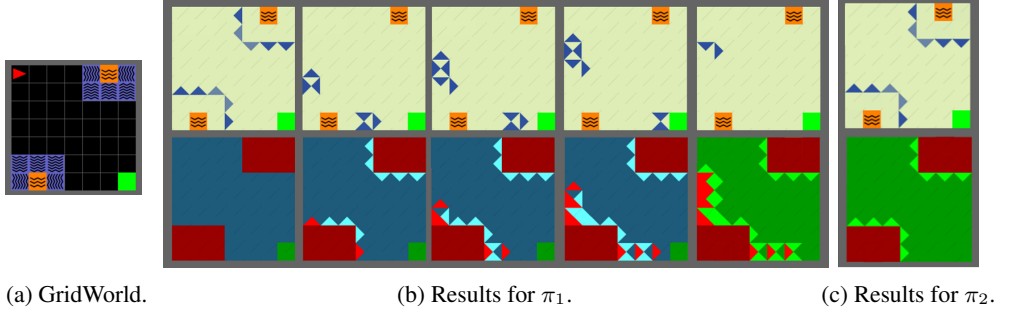

(a) GridWorld.  (b) Results for $\pi_1$.  (c) Results for $\pi_2$.

Figure 2: Slippery Gridworld example: setting (left), visualization of evaluating $\pi_1$ (middle), and $\pi_2$ (right).

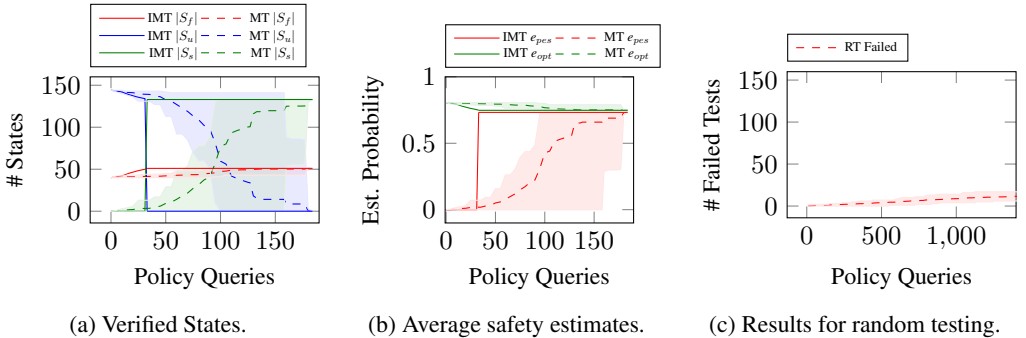

(a) Verified States.  (b) Average safety estimates.  (c) Results for random testing.

Figure 3: Slippery Gridworld example: Evaluation results of $\pi_1$.

## 4 Experimental Evaluation

All details of the experimental setup can be found in Appendix A. We provide the implementation and tested policies as supplementary material. We compare IMT with our model-based approach *without* importance ranking (MT) and model-free random testing (RT) as a baseline. Thus, in MT, our algorithm restricts the MDP by the sampled agent's decisions and computes $e_{opt}$ and $e_{pes}$ to provide evaluation results on the entire state space but *samples the policy randomly*. For RT, the policy is executed from random states for a certain number of steps. Any violation of the evaluation objective is reported. We report the runtimes averaged over 10 runs for each experiment in seconds, unless indicated otherwise, as *total time($\pm$ STDev) / total time for computing the estimates($\pm$ STDev) / total time for querying the policy ($\pm$ STDev)*.

### 4.1 Slippery Gridworld

We performed our first experiment in the Farama Minigrid environment [27]. A description of the environment and the RL training parameters are given in Appendix B. The Gridworld is depicted in Fig. 2a. The agent has to reach the green goal without touching the lava. The lava is surrounded by slippery tiles, stepping on which carries a predefined probability of slipping into lava. The size of the state space is $|\mathcal{S}| = 7 \times 7 \times 4 = 196$, with 49 cells multiplied by 4 for the different orientations of the agent. The safety objective $\varphi$ requires the agent to not enter the lava with a probability $\delta_\varphi \geq 1.0$.

**RL training parameters.** We trained policies $\pi_1$ and $\pi_2$ by utilizing a DQN. We used a sparse reward function with a reward of 1 for reaching the green goal and $-1$ for falling into the lava. We trained $\pi_1$ using a fixed initial state and $\pi_2$ using initial states uniformly sampled from $\mathcal{S}$.

**IMT/MT parameters.** We used a horizon of $n = \infty$, a minimal difference of $\varepsilon_\varphi = 0.05$, a number of samples per iteration of $m = 10$. For IMT, no states with a ranking value close to 0 were sampled.

**Visualizing IMT.** Fig. 2b visualizes the iterations of our IMT algorithm when evaluating $\pi_1$. Per iteration, the picture on the top visualizes the highest-ranked states, with the intensity of the color capturing the ranking. Note that a state represents the $(x, y)$-coordinates of the grid and the orientation of the agent and is thus visualized as a triangle. Per iteration, IMT samples $\pi_1$ in the highest-ranked

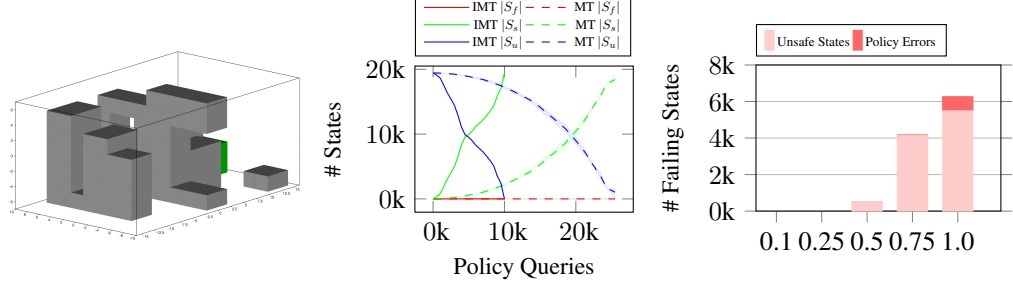

(a) UAV Reach-Avoid setting.      (b) Results for $noise = 0.1$      (c) Number of safety violations.

Figure 4: UAV Task: setting (4a), verified states (4b), and number of identified safety violations (4c).

states (blue triangles) and computes the estimates. The pictures on the bottom show the updated sets of verified states after computing the estimates: $\mathcal{S}_s$ is visualized in green, $\mathcal{S}_f$ in red, and $\mathcal{S}_u$ in blue. Brighter colors represent states in which the decisions of $\pi_1$ were sampled. IMT terminates after 5 iterations when evaluating $\pi_1$. Note that the evaluation iteratively reveals the area in which $\pi_1$ violates safety. Fig. 2c visualizes the evaluation of the policy $\pi_2$. IMT terminates after a single iteration and positively verifies $\pi_2$ in all states in which the safety objective can be fulfilled.

**Evaluation results.** Fig. 3a plots the number of verified states when evaluating $\pi_1$. Solid lines represent IMT results, dashed lines represent MT, where green lines represent $|S_s|$, red lines represent $|S_f|$, and blue lines $|S_u|$. We repeated the analysis via MT 10 times: the shaded area represents the minimal and maximal values, and the dashed lines the average number of states. After sampling $\pi_1$ only 33 times, IMT terminates with $|\mathcal{S}_s| = 145$, $|\mathcal{S}_f| = 51$, and $|\mathcal{S}_u| = 0$. Thus, IMT provides complete verification results of $\pi_1$ over the entire state space with only 33 policy samples. In contrast, on average, MT verifies the entire state space after sampling the agent's decisions almost on the entire state space. Fig. 3b plots the values for the optimistic (green) and pessimistic (red) safety estimates for IMT (solid lines) and MT (dashed lines), averaged over all states, which show that the averaged estimates of IMT tighten faster than for MT. Finally, we report the findings of RT in Fig. 3c when executing a test case for 10 steps. The results show the clear advantage of exploiting our testing approach. By utilizing testing with model checking, we obtain verification results on the entire state space in contrast to RT which is only able to report a small number of states from which $\varphi$ is violated.

**Runtimes.** The costs for computing the estimates per iteration are in the range of milliseconds. The total runtime to verify $\pi_1$ was $12.29(\pm 0.7)$ / $1.11(\pm 0.10)$ / $0.11(\pm 0.01)$, with IMT and $25.62(\pm 1.8)$ / $3.21(\pm 0.23)$ / $0.41(\pm 0.02)$ with MT.

## 4.2 UAV Reach-Avoid Task

For the second set of experiments, we test policies computed for drone navigation by Badings et al. [28]. We refer to this work for details regarding the policy and environment, which is illustrated in. Fig. 4a. The task of the drone is to navigate to the goal location (green box). The safety objective $\varphi$ states that the drone must not collide with a building (grey boxes) and must stay within the boundaries with a probability $\delta_\varphi \geq 0.95$. The state space $|\mathcal{S}|$ comprises $25.517$ states. The wind in the simulation affects the drone, which is modeled stochastically and controlled through the parameter $\eta$.

**IMT parameters.** We used $m = 500$ samples per iteration, $n$, and $\varepsilon_\varphi$, as above.

**Evaluation results.** Our testing approach IMT/MT was able to verify control policies computed under five difference noise settings of $\eta \in \{0.1, 0.25, 0.5, 0.75, 1.0\}$ over the entire state space. Fig. 4c gives the number of verified unsafe states per policy. All policies with $\eta < 0.75$ are verified safe in all states from which it is possible to behave safely (light red bars indicate states from which safety violations cannot be avoided). Even though the policies have been specially designed to be safe, IMT was able to find safety violations for policies with $\eta \geq 0.75$. The policies showed unsafe behavior in 15 or 775 additional states, respectively, for which safe behavior would have been possible (dark bars). Fig. 4b shows the verification results for IMT and MT for $\eta = 0.1$. The test results for the policies with $\eta \geq 0.25$ can be found in Appendix C. As before, adding importance-ranking for sampling the policy decreases the number of required samples to verify the policy. We performed RT

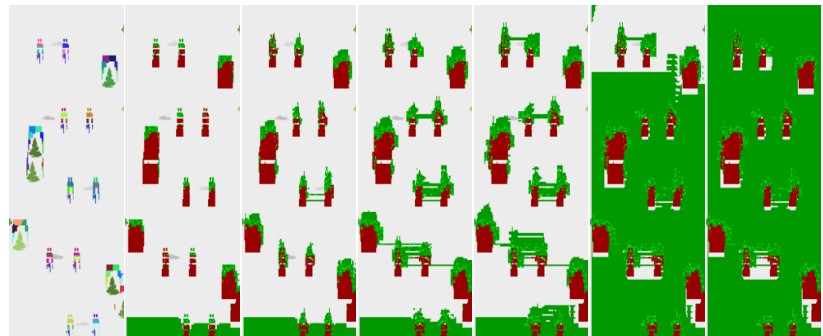

Figure 5: The initial clustering and iterations of the algorithm for an average cluster size of 25.

with a budget of 50.000 queries and a maximum number of 3 time steps per test case. Averaged over 10 runs, RT found 1120 ($\pm$76.29) failed test cases for the policy with $\eta = 1.0$ and 613 ($\pm$53) failed test cases for $\eta = 0.75$.

**Runtimes.** The runtimes for evaluating the policy are $269.8(\pm5.5)/ 73.74(\pm1.1)/ 0.02(\pm0.02)$ for IMT, and $1793(\pm21.1) / 185.28(\pm3.2) / 0.02(\pm0.02)$ for MT for a noise level of $\eta = 0.2515$ This shows that for larger examples, adding importance-based sampling significantly reduces the time needed to verify the policy.

### 4.3 Atari Skiing

We have evaluated IMT with clustering, IMTc for short, by testing a learned policy for Atari Skiing [29]. In Skiing, the player controls the tilt of the skies to reach the goal as fast as possible. The safety objective $\varphi$ is to avoid collisions with trees and poles with a probability of $\delta_\varphi \geq 1.0$. A state describes the $(x, y)$ position, the $tilt \in [1..8]$ of the ski, and velocity $v \in [0..5]$ of the skier. The state space $\mathcal{S}$ comprises roughly $2.2 * 10^6$ states.

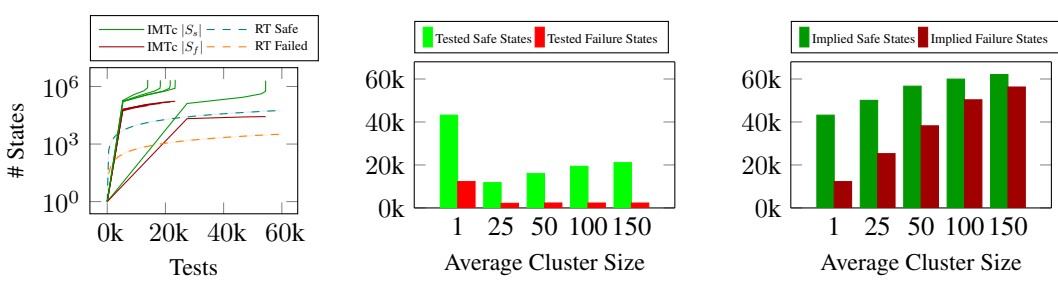

(a) Verified States.  (b) Number of safe and failed tests.  (c) Number of states in the final clusters.

Figure 6: Atari Skiing Example: Evaluation results for the tested policy.

**IMT parameters.** We used a time horizon of $n = 200$, a minimal difference of $\epsilon_\varphi = 0.05$, and a fraction $\kappa = 0.2$ of tested states per cluster. The clusters have been computed using $k$-means for states with $\delta_i > 0.8$ with a $k$ to create average cluster sizes of $\zeta \in \{25, 50, 100, 150\}$.

**Visualizing IMT.** Fig. 5 visualizes the initial clustering of the highest-ranked states and the iterations of IMTc with an average cluster size of $\zeta = 25$. We show the results for states in which $tilt = 4$, i.e. the skier is *aligned with the slope*, and $v = 4$. The visualization for different values of $tilt$ and $v$ can be found in Appendix E. We depict states from which the policy has been tested with lighter colors. Darker colors depict implied results.The results show that the agent robustly learned to avoid collisions (it avoids any collision as long as it is not placed too close to an obstacle).

**Evaluation results.** We evaluated IMTc using different values for $\zeta$ and compared it with IMT, i.e. $\zeta = 1$, and RT. Fig. 6a plots the total number of failure states $\mathcal{S}_f$ and safe states $\mathcal{S}_s$ for the whole state space over the number of executed test cases for different values of $\zeta$. The green curves (left to right) plot the results for $\mathcal{S}_s$ using the cluster sizes $\{25, 50, 100, 150, 1\}$, the red curves for $\mathcal{S}_f$ accordingly. For comparison, we executed RT 10 times and plot the average number of failing (orange dashed) and safe test cases (teal dashed), where the shaded areas show the minimal and

maximal values. The results show that IMTc terminates faster with smaller cluster sizes. Larger cluster sizes overapproximate unsafe regions more heavily. Thus, more testing effort is needed around the unsafe regions in the subsequent iterations. However, all instances of IMTc reduce the testing budget required compared to IMT, which was to be expected since only $20\%$ of the states of each cluster were tested. These facts are also underlined by Figures 6b and 6c, which show the number of safe and failed tests, and the implied verdicts for cluster states in the final iteration, respectively. Fig. 6b shows that clustering heavily increases the scalability of our approach since it lowers the needed testing budget by up to a factor of 5 for $\zeta = 25$.

**Runtimes.** The runtimes for evaluating the policy, excluding the time needed to render the testing results, for $\zeta \in \{25, 50, 100, 150\}$ are 86 minutes ($\pm 8.3$) / 40 ($\pm 4.5$) / 8.5 ($\pm 2.3$). Evaluating the policy for $\zeta = 1$ took 127 minutes / 59 ($\pm 7.3$) / 35.9 ($\pm 4.4$).

## 5    Conclusion & Future Work

We presented importance-driven testing for RL agents. The process iteratively (1) ranks the states based on the influence of the agent's decisions on the expected overall safety, (2) samples the DRL policy under test from the ranking, and (3) restricts the model of the environment. By utilizing probabilistic model checking, our algorithm provides upper and lower bounds on the expected outcomes of the policy execution across all modeled states in the state space. These estimates provide formal guarantees about the violation or compliance of the policy to formal properties. We presented an extension of the basic algorithm by introducing clustering to increase scalability. In future work, we will adapt IMT to allow the testing of stochastic policies by adapting the restriction of the MDP and the verification procedure. We will Furthermore, we will introduce several abstraction techniques to further increase the scalability of our approach. Finally, we will use recently proposed approaches to both learn discrete models of domains that are continuous in both their state and action spaces to increase the applicability of IMT and learn the MDP online during the training phase of the policy.

Bettina Könighofer and Stefan Pranger were supported by the State Government of Styria, Austria - Department Zukunftsfonds Steiermark, Martin Tappler was supported by the WWTF project ICT22-023, and Hana Chockler was supported in part by the UKRI Trustworthy Autonomous Systems Hub (EP/V00784X/1), the UKRI Strategic Priorities Fund to the UKRI Research Node on Trustworthy Autonomous Systems Governance and Regulation (EP/V026607/1), and CHAI - EPSRC Hub for Causality in Healthcare AI with Real Data (EP/Y028856/1). We thank both Antonia Hafner and Martin Plank for their proof-of-concept implementations of the experimental evaluation.

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

## A  General Experimential Setup

All experiments have been executed on a desktop computer with a $8 \times 3.9$GHz Intel i5-8265U CPU and 16GB of RAM using a *single worker*, i.e. we did not use any form of multithreading.

We implemented our importance-driven testing framework in `Python` and used the probabilistic model checking tool `Tempest` [30] to compute the estimates and the importance ranking.

## B  Details for the Gridworlds Experiments.

In this section, we give the details of the experiments in Section 4.1 and Appendix D.2.

**Environmental Details.** The agent behaves in the style of an omnidirectional robot. It is able to perform seven actions: Moving forward, turning left, turning right, picking up objects, dropping the object being carried, interacting with doors or other objects, and idlying. The slippery tiles in 2a introduce stochastic behaviour. If the agent tries to move forward on a slippery tile, it only manages to move to its intended tile in front of it with a probability of $\frac{3}{9}$. Otherwise, it slips

- with a probability of $\frac{1}{9}$ to either the adjacent tile to its left or its right, respectively, or
- with a probability of $\frac{2}{9}$ to either the tile to the left or to the right of the tile in front of the agent, respectively.

The tiles belonging to one-way streets in 8a, depicted by a blue arrow, do not allow the agent to move against the direction of the one-way. This especially means that an agent is not allowed to enter a one-way from the wrong side.

**RL Training Details.** We used a standard implementation of DQN from *stable-baselines3* [31] with a CnnPolicy. The network to classify and train the agent follows a standard approach taken from [32]: It features 3 convolutional layers and a linear activation layer. The learning parameters have been slightly altered with the following modifications:

- *discount factor* $\gamma$: 0.95
- *exploration scheme:* A linear decay from 0.7 to 0.01 over the first 90% of the learning duration.

The agents for policies $\pi_1$, $\pi_2$, $\pi_3$, and $\pi_4$ have been trained with a total number of 500000 steps. An episode lasted a maximum number of 100 timesteps or ended prematurely if the agent caused a safety violation or if it reached the goal.

# C Additional Results for UAV Reach-Avoid Experiment

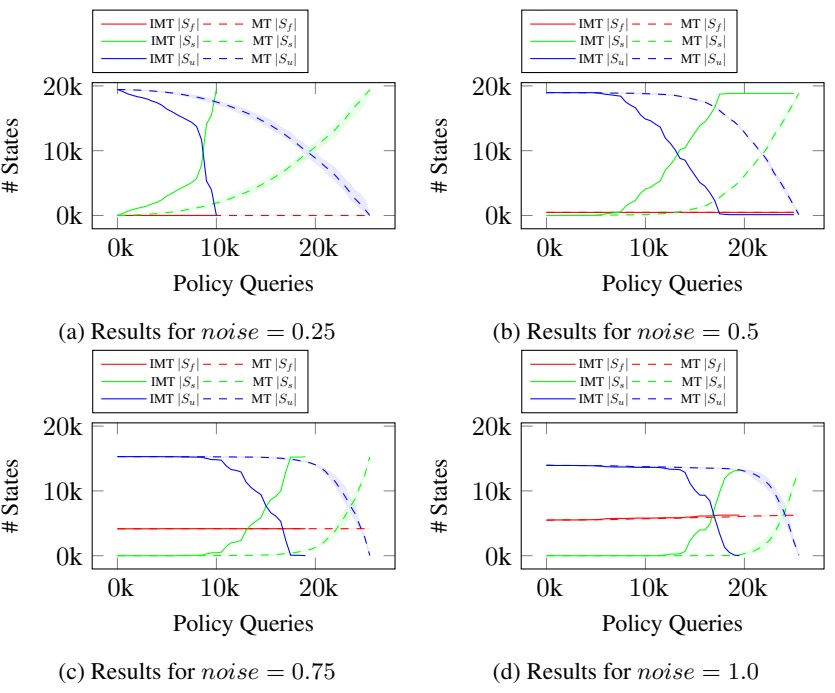

(a) Results for $noise = 0.25$

(b) Results for $noise = 0.5$

(c) Results for $noise = 0.75$

(d) Results for $noise = 1.0$

Figure 7: Evaluation results for the UAV Reach-Avoid task under noise levels $0.25, 0.5, 0.75$ and $1.0$.

| $\eta$ | 0.1 | 0.25 | 0.5 |
|---|---|---|---|
| IMT | 269sec. ($\pm 5.5$) | 320sec. ($\pm 16.1$) | 1264sec. ($\pm 3.8$) |
| MT | 1793sec. ($\pm 21.7$) | 2227sec. ($\pm 0.2$) | 2570sec. ($\pm 125.4$) |

| $\eta$ | 0.75 | 1.0 |
|---|---|---|
| IMT | 1383sec. ($\pm 4.4$) | 2422sec. ($\pm 3.87$) |
| MT | 2844sec. ($\pm 34.3$) | 4016sec. ($\pm 6.23$) |

Table 1: Average synthesis times for the different policies.

# D Testing for Performance

In this section we discuss the necessary background and definitions needed to adapt IMT for testing for performance.

**Model checking of performance objectives.** Model checking can be used to compute the expected accumulated reward for all states and actions in $\mathcal{M}$. In particular, for a given MDP $\mathcal{M}$, a policy $\pi$, and a reward function $\mathcal{R} : \mathcal{S} \to \mathbb{R}$, it computes the following values:

- $\mathbb{E}_{\mathcal{M}^\pi, \mathcal{R}} : \mathcal{S} \times \mathbb{N} \to \mathbb{R}$ gives the expected accumulated reward in $\mathcal{M}^\pi$ from a state $s$ within $n$ steps.
- $\mathbb{E}_{\mathcal{M}, \mathcal{R}}^{\max}(s, n) = \max_{\pi \in \Pi} \mathbb{E}_{\mathcal{M}^\pi, \mathcal{R}}(s, n)$ gives the *maximal* expected accumulated reward *over all policies in* $\Pi$ from a state $s$ within $n$ steps.
- $\mathbb{E}_{\mathcal{M}, \mathcal{R}}^{\min}(s, n) = \min_{\pi \in \Pi} \mathbb{E}_{\mathcal{M}^\pi, \mathcal{R}}(s, n)$ gives the *minimal* expected accumulated reward *over all policies in* $\Pi$ from a state $s$ within $n$ steps.

A *performance objective* $\langle \mathcal{R}, \delta_{\mathcal{R}} \rangle$ is defined over the reward function $\mathcal{R}$ and a threshold $\delta_{\mathcal{R}} \in \mathbb{R}$ that defines the lowest-acceptable expected accumulated reward over $n$ steps.

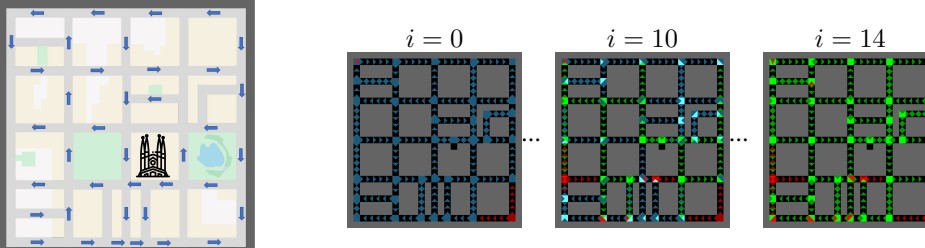

(a) Map of an area in Barcelona.

(b) Selected results for iterations $i = 0$, $i = 10$, $i = 14$ of testing $\pi_3$.

Figure 8: Urban navigation example: setting (left) and visualization of evaluating $\pi_3$ (right).

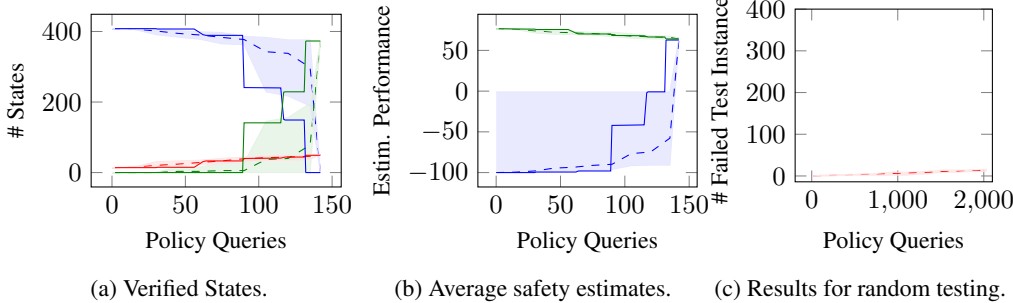

(a) Verified States.

(b) Average safety estimates.

(c) Results for random testing.

Figure 9: Urban navigation example: Evaluation results of $\pi_3$.

**Definition D.1** (Performance objective). Given an MDP $\mathcal{M} = (\mathcal{S}, \mathcal{A}, \mathcal{P}, \mu)$, a reward function $\mathcal{R} : \mathcal{S} \to \mathbb{R}$, and a threshold $\delta_{\mathcal{R}} \in \mathbb{R}$. A policy $\pi$ satisfies the performance objective $\langle \mathcal{R}, \delta_{\mathcal{R}} \rangle$ from a given state $s \in \mathcal{S}$ within a given number of steps $n$ if

$$\mathbb{E}_{\mathcal{M}^\pi, \mathcal{R}}(s, n) \geq \delta_{\mathcal{R}}.$$

### D.1 Importance-driven Performance Testing

IMT can be easily adapted to evaluate a policy $\pi$ for performance objectives. To tailor Alg. 1 for performance testing, we provide as inputs a performance objective $\langle \mathcal{R}, \delta_{\mathcal{R}} \rangle$, and a minimal difference in performance $\varepsilon_{\mathcal{R}}$ between optimistic and pessimistic estimates. These inputs replace the corresponding safety-related parameters $\varphi$, $\delta_\varphi$, and $\varepsilon_\varphi$. The performance estimates (Line 3) are defined by:

**Definition D.2** (Performance estimates). For a given MDP $\mathcal{M} = (\mathcal{S}, \mathcal{A}, \mathcal{P}, \mu)$, a reward function $\mathcal{R} : \mathcal{S} \to \mathbb{R}$, and a given number of $n$ steps, the *optimistic* and *pessimistic performance estimate* $e_{opt, \mathcal{R}}, e_{pes, \mathcal{R}} : \mathcal{S} \times \mathbb{N} \to \mathbb{R}$ are defined as follows:

$$\forall s \in \mathcal{S}: e_{opt, \mathcal{R}}(s, n) = \mathbb{E}_{\mathcal{M}, \mathcal{R}}^{\mathsf{max}}(s, n), \text{ and } \forall s \in \mathcal{S}: e_{pes, \mathcal{R}}(s, n) = \mathbb{E}_{\mathcal{M}, \mathcal{R}}^{\mathsf{min}}(s, n).$$

For the optimistic performance estimate, the computation assumes that the policy $\pi$ selects the action that maximizes the expected reward in each unrestricted state. Conversely, for the pessimistic estimate, the assumption is that the least optimal actions concerning the reward are chosen. A state $s \in \mathcal{S}$ satisfies the performance objective $\langle \mathcal{R}, \delta_{\mathcal{R}} \rangle$ if $e_{pes, \mathcal{R}}(s, n) \geq \delta_{\mathcal{R}}$ (Line 4). A state $s \in \mathcal{S}$ violates the performance objective if $e_{opt, \mathcal{R}}(s, n) \leq \delta_{\mathcal{R}}$ (Line 5).

For the stopping criterion, the difference between performance estimates is compared to $\varepsilon_{\mathcal{R}}$ (Line 7).

### D.2 Urban Navigation Task

We modeled a part of Barcelona in a Minigrid environment as illustrated in Fig. 8a. The size of the state space is $|\mathcal{S}| = 426$. The agent's task is to navigate to Sagrada Família (within 100 steps), while respecting the traffic rules, i.e., not driving against the one-ways. We trained and evaluated two policies $\pi_3$ and $\pi_4$.

**RL training parameters.** $\pi_3$ and $\pi_4$ were trained utilizing a DQN. The agent is given the reward of 1 for reaching the goal and additionally $-0.01$ per step. Both policies have been trained using initial states uniformly sampled from $|\mathcal{S}|$.

**IMT/MT parameters.** We used the same parameters as for the experiment in Section 4.1, but used $m = 15$.

**Visualizing IMT.** Fig. 8b visualizes the sets of verified states for $\pi_3$ in selected iterations of Alg. 1. We visualize $\mathcal{S}_s$, $\mathcal{S}_f$, and $\mathcal{S}_u$ in the same way as in our first experiment. Brighter colors again represent states where $\pi_3$ was sampled. IMT iteratively samples the agent's decisions at crossings ranked on the difference the decisions of the individual roads have on the total length of the path to Sagrada Família. Figures 10 and 11 visualize $\mathcal{S}_s$, $\mathcal{S}_f$, and $\mathcal{S}_u$ for $\pi_4$. IMT needs 9 iterations to fully verify that $\pi_4$ behaves optimally in any of the modelled states.

**Evaluation results.** Fig. 9a plots the number of verified states and Fig. 9b plots the estimates when evaluating $\pi_3$. As above, we compare IMT and the average results for MT over 10 runs. Even though $\pi_3$ performed well on large parts of the state space, IMT and MT identified wrong decisions at several crossings that do not allow the agent to reach the goal in time. For RT we executed $\pi_3$ with a budget of 2000 policy queries and a maximum number of 100 time steps. Fig. 9c plots the average number of identified violations. While IMT and MT only have to sample the agent's decisions at the crossings to verify the entire state space, RT on average only found $13.25$ ($\pm 2.75$) states from which a test case failed. Fig. 12a plots the number of verified states and 12b plots the estimates when evaluating $\pi_4$.

**Runtimes.** The runtime to verify $\pi_3$ was 36.79 sec ($\pm 2.0$) with MT and 24.70 sec ($\pm 2.2$) with IMT. Both approaches MT and IMT needed a similar total runtime of about 9.84sec. ($\pm 0.2$) to verify $\pi_4$.

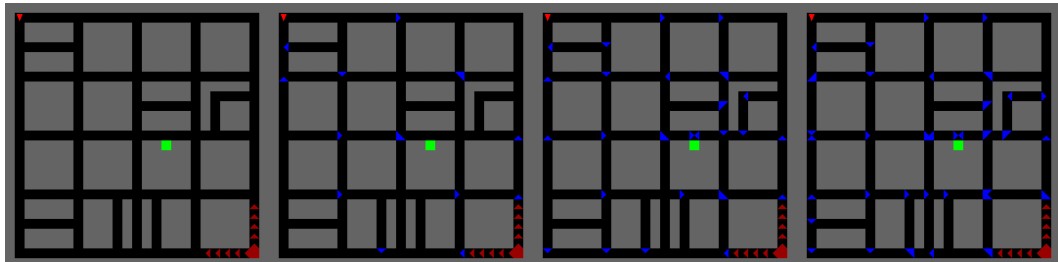

Figure 10: The intermediate results for iteration $0 - 4$ for the verification of $\pi_4$ using IMT.

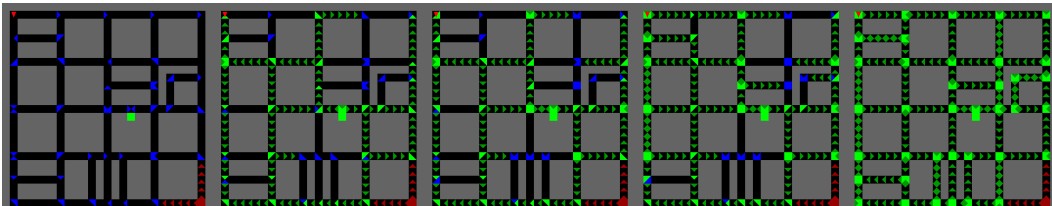

Figure 11: The intermediate results for iteration $5 - 9$ for the verification of $\pi_4$ using IMT.

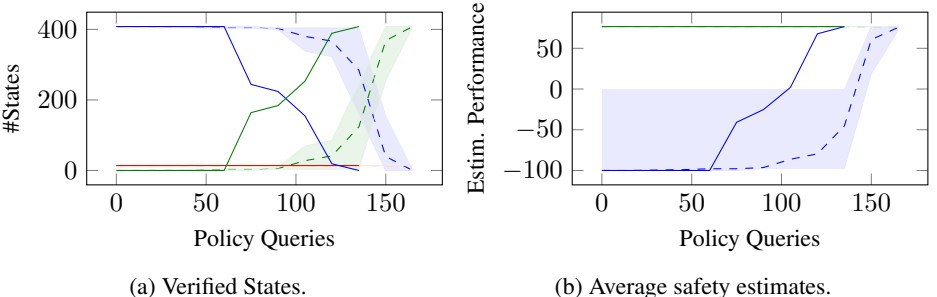

(a) Verified States.

(b) Average safety estimates.

Figure 12: Urban navigation example: Evaluation results of $\pi_4$.

# E    Additional Results for Atari Skiing Experiment

## E.1    Results for $\zeta = 25$

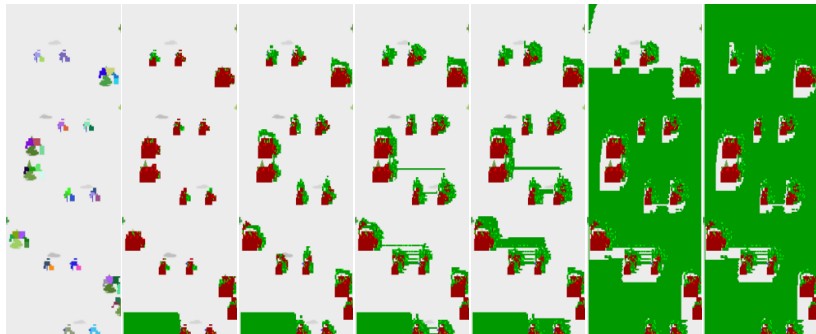

Figure 13: The initial clustering and iterations of IMT for an average cluster size of 25, $tilt = 2$, and $v = 2$.

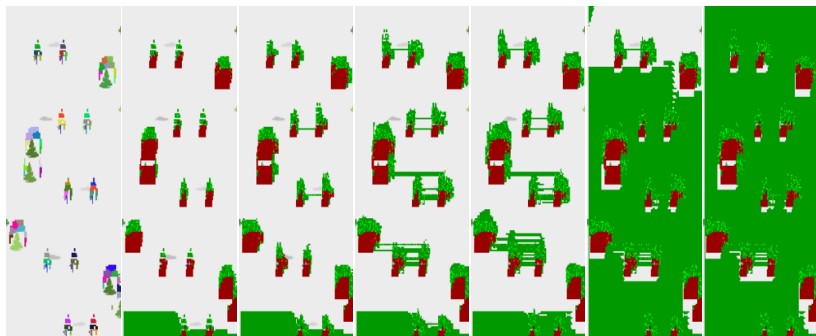

Figure 14: The initial clustering and iterations of IMT for an average cluster size of 25, $tilt = 4$, and $v = 4$.

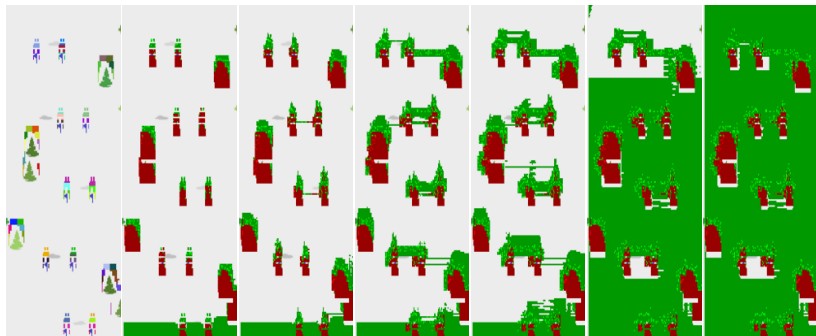

Figure 15: The initial clustering and iterations of IMT for an average cluster size of 25, $tilt = 5$, and $v = 4$.

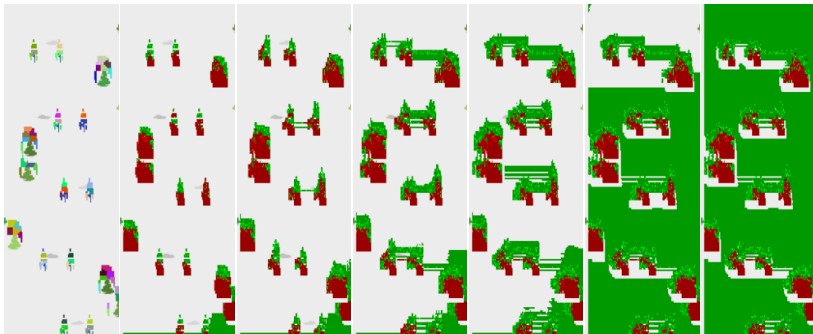

Figure 16: The initial clustering and iterations of IMT for an average cluster size of 25, $tilt = 6$, and $v = 3$.

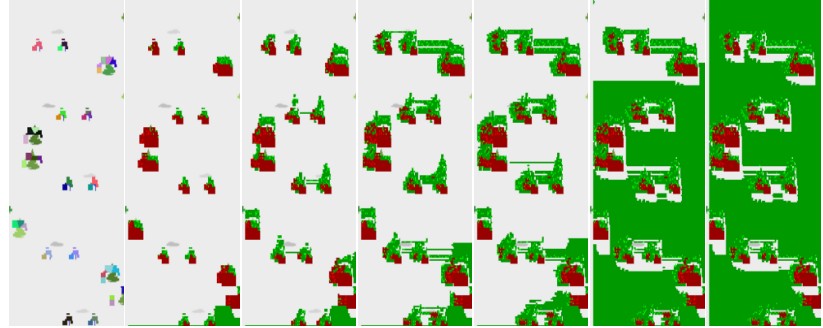

Figure 17: The initial clustering and iterations of IMT for an average cluster size of 25, $tilt = 7$, and $v = 2$.

## E.2  Results for $\zeta = 100$

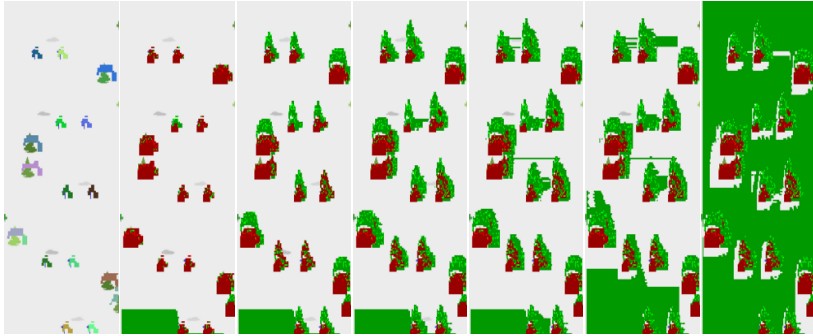

Figure 18: The initial clustering and iterations of IMT for an average cluster size of 100, $tilt = 2$, and $v = 2$.

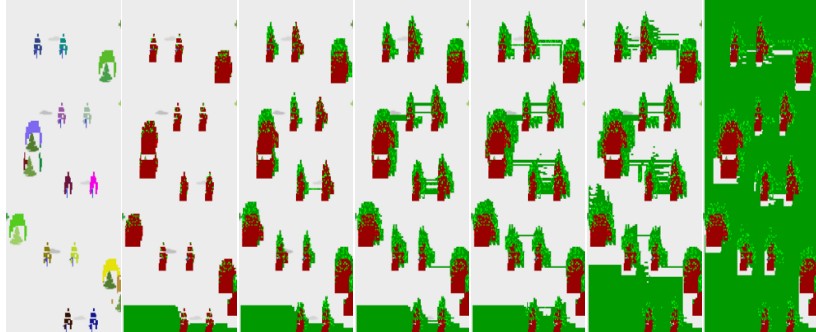

Figure 19: The initial clustering and iterations of IMT for an average cluster size of 100, $tilt = 4$, and $v = 4$.

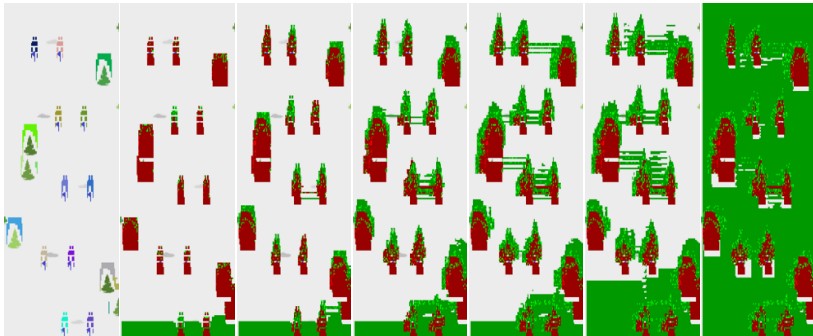

Figure 20: The initial clustering and iterations of IMT for an average cluster size of 100, $tilt = 5$, and $v = 4$.

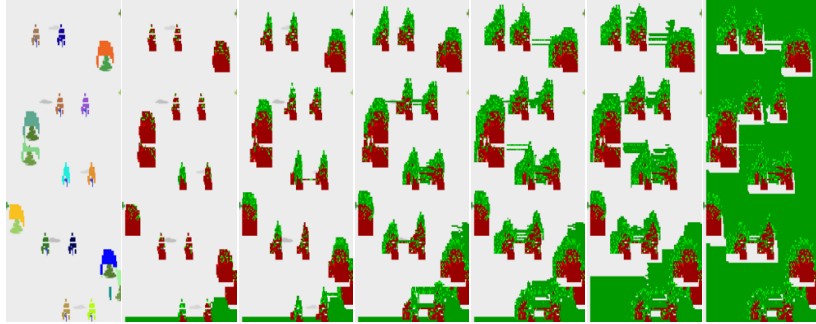

Figure 21: The initial clustering and iterations of IMT for an average cluster size of 100, $tilt = 6$, and $v = 3$.

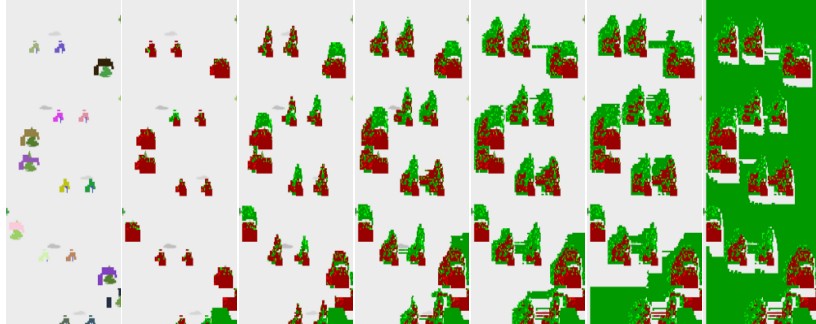

Figure 22: The initial clustering and iterations of IMT for an average cluster size of 100, $tilt = 7$, and $v = 2$.

