# OpenReview forum: "Test Where Decisions Matter: Importance-driven Testing for Deep Reinforcement Learning"
_NeurIPS.cc/2024/Conference — NeurIPS 2024 poster_

### Official Review · Reviewer_4erK · 2024-07-08

**Soundness:** 3
**Presentation:** 3
**Contribution:** 3
**Rating:** 7
**Confidence:** 5

**Summary:**

This paper proposes a model-based method for testing the safety properties of Deep Reinforcement Learning (DRL). The method computes a ranking of state importance across the entire state space, dividing the state space into safe and unsafe regions. The approach provides optimal test-case selection and guaranteed safety by providing formal verification guarantees over the entire state space. The method is evaluated on several examples, showing it discovers unsafe policy behavior with low testing effort.

**Strengths:**

+ The paper introduces a novel method to rank state importance across the entire state space, which further assist the testing procedure.
+ The use of optimistic and pessimistic safety estimates provides a range of expected outcomes and is interesting
+ The authors conduct a detailed evaluation of the method on several popular DRL tasks and demonstrate its effectiveness.

**Weaknesses:**

- Some technical details should be explained more clearly.
- The authors doesn't compare IMT with the SOTA DRL testing method, like [15] mentioned by the authors in the paper.

**Questions:**

In general, the idea of ranking the importance of the state and leveraging it as the testing guidance is very interesting. I believe this work would have a certain impact on the further development of DRL testing.

There are a few technical points unclear.

1. Page 2, Line 60. It would be good if the authors gave a more concrete concept of the formal guarantee provided by IMT, like probabilistic certification.

2. Page 5, Line 175. What do you mean by "significantly larger"? Do you perform statistical testing here?

Moreover, some related work is missing:

- Gerasimou, Simos, Hasan Ferit Eniser, Alper Sen, and Alper Cakan. "Importance-driven deep learning system testing." In Proceedings of the ACM/IEEE 42nd International Conference on Software Engineering, pp. 702-713. 2020.
- Song, Jiayang, Xuan Xie, and Lei Ma. "$\mathtt {SIEGE} $: A Semantics-Guided Safety Enhancement Framework for AI-enabled Cyber-Physical Systems." IEEE Transactions on Software Engineering (2023).
- Shi, Ying, Beibei Yin, and Zheng Zheng. "Multi-granularity coverage criteria for deep reinforcement learning systems." Journal of Systems and Software 212 (2024): 112016.

**Limitations:**

Yes, the authors address it in the guideline.

---

> ### Author Rebuttal · Authors · 2024-08-07
>
> We thank the reviewer for finding our idea of guiding DRL testing through an importance ranking interesting. Given the current lack of testing methodologies for DRL and the numerous potential extensions of our approach, we believe our method could significantly impact the development of testing strategies for DRL. We will address the identified weaknesses, especially we will clarify the technical details. Please also see the global rebuttal, where we address the comparison to other baselines.
>
> ## Q1: Concept of formal guarantees
>
> We agree that the provided formal guarantees should already be stated clearly in the introduction. We will amend this in the introduction.
>
> In Section 3.1. (Lines 149-158), we explain the provided formal guarantees:
>
> The estimates provide lower and upper bounds on the expected outcomes of the policy execution across all modeled states in the state space.
> If the optimistic estimate violates a safety objective, then the policy is guaranteed to violate the safety objective, i.e., the probability that the safety CTL formula is satisfied using the agent’s policy is lower than the defined safety threshold.
> On the other hand, any safety objective assured by the pessimistic estimate are formally proven to hold for the policy, i.e., the probability that the safety CTL formula is satisfied using the agent’s policy is greater or equal than the defined safety threshold.
>
> We will give a similar explanation in the introduction.
>
> ## Q2: “significant” difference (Line 175). Are you performing statistical testing?
>
> The values for $e_{opt}(s,a,n)$ and $e_{opt}(s,a',n)$ are directly computed using sound value iteration which yields exact probabilities. Thus, we are not performing statistical testing.
>
> We consider a state important if the maximal difference for two distinct actions is substantially larger than 0, i.e. if the decision of the agent has an important impact on the expected probability to ensure safety.  We apologize for the misleading phrasing. We will rephrase this and clarify it in the camera-ready version.
>
> ## Other Comments - Related Work
>
> We thank the reviewer for giving us pointers to related work. We will include all three references in the related work section.
> Highly relevant are [1] and [3]. [1] defines importance over the most-important neurons and tests the behavior of the most important neurons. In contrast, we define the ranking over the environment states, from which the policy should be explored. [3] is relevant since they are the first that discuss coverage criteria for DRL testing.
> Additionally to comparing to other testing frameworks for DRL, we compare with approaches that combine model-based formal methods with model-free RL. [2] is clearly relevant in that regard, as they are combining DRL and model learning to design controllers that satisfy STL properties.
>
> [1] Gerasimou, Simos, Hasan Ferit Eniser, Alper Sen, and Alper Cakan. "Importance-driven deep learning system testing."
>
> [2] Song, Jiayang, Xuan Xie, and Lei Ma. "SIEGE: A Semantics-Guided Safety Enhancement Framework for AI-enabld Cyber-Physical Systems."
>
> [3] Shi, Ying, Beibei Yin, and Zheng Zheng. "Multi-granularity coverage criteria for deep reinforcement learning systems."

---

> > ### Comment · Reviewer_4erK · 2024-08-14
> >
> > Thank you for the rebuttal! It address my concerns and I think it should be accepted.

---

### Official Review · Reviewer_xBae · 2024-07-10

**Soundness:** 2
**Presentation:** 3
**Contribution:** 2
**Rating:** 5
**Confidence:** 3

**Summary:**

This paper proposed a novel model-based importance-driven framework for testing trained DRL policies. By focusing on testing the cases with higher importance ranks, the proposed framework improves the testing scalability. The evaluation of several case studies demonstrates that the proposed framework can more efficiently discover unsafe policy behavior with low testing effort.

**Strengths:**

1. The paper is well-structured and easy to follow.
2. The idea of testing on important samples is interesting.

**Weaknesses:**

1. The definition of the safety formula $\phi$  and the formula for calculating $\mathbb{P}\_{\mathcal{M}^\pi, \phi}$ should be stated clearly.
2. The current framework depends on several human-defined thresholds, for instance, $\delta_{\phi}$, $\epsilon_{\phi}$ $\delta_{i}$. How those parameters are determined and sensitive to the final testing result should be discussed more clearly.
3. The current framework is shown to be applicable in a discrete state and action space. It would be nice to see the discussion on the potentials and limitations applied to the continuous state and action space.

**Questions:**

1. The importance ranking is based on the maximal difference between the optimistic estimates concerning the available actions, which considers the impact of decisions on the optimistic estimates. Why not use the pessimistic estimate, as it is more concerned about safety violations?

---

> ### Author Rebuttal · Authors · 2024-08-07
>
> We thank the reviewer for finding our testing approach interesting. We believe our method could significantly impact the development of testing strategies for DRL and hope that it will lead to interesting follow up research. We will answer the questions and address the mentioned weaknesses in the following.
>
> ## Q1. Why is the importance ranking based on the maximal difference between the optimistic estimates? Why not use the pessimistic estimate, as it is more concerned about safety violations?
> To evaluate the safety of executing action a1​ in state s0​, we must assess the safety of any state s1​ that can be reached from s0​ via a1​. To avoid sampling the agent’s policy in s1​ and subsequent states, we can either assume the agent follows the safest policy (using $P_{max}$​) or the least safe policy (using $P_{min}$​). Here, we justify why our testing framework uses $P_{max}$​ ($e_{opt}$) rather than $P_{min}$ ($e_{pes}$) to assess safety.
>
> Using $e_{opt}$:
> If for a given state s0​, there exist actions a1 and a2 with a large difference in the maximal expected probability of satisfying $\varphi$, this state will be highly ranked.
>
> Suppose we want to evaluate how safe it was to take action a1 in s0. Assume the transition (s0,a1,s1) occurs with high probability and s1​ has a high optimistic estimate. In this case, there exists a safe policy from s1​ (which the agent could choose to follow). Thus, entering s1​ was not safety-critical since any safety issues could be avoided from s1​.
>
> Further, assume that if the agent selects action a2​ from s0​, the transition (s0,a2,s2) occurs with high probability and s2 has a low optimistic estimate (i.e., it is not possible to execute a safe policy from s2 anymore). In this scenario, state s0 is highly ranked as it is crucial to choose action a1 over a2.
>
> Using $e_{pes}$:
> The pessimistic estimate does not have this property since it is based on the minimal expected probability of satisfying $\varphi$. Due to the possibility of reaching unsafe states from most states, the difference between actions a1​ and a2​ does not reflect the impact of the action choice on mitigating a safety violation as the optimistic estimate does. This leads to similar rankings for most states.
>
> Consequently, we would need to test any state from which the agent could cause a safety violation. For example, in a skiing scenario, even if the agent is far from any object, we would still need to test the state even if from any successor state, crashes can still easily be avoided. This is exactly what we want to avoid.
>
> Once we have sufficiently restricted the environment MDP, the pessimistic estimate becomes more informative. It can then identify candidate states that have not been tested and from which unsafe areas cannot be entered. This is highlighted in lines 151 to 155.
>
> ## W1: Clearly state the safety formula $\varphi$ and the formula for calculating the probabilities
> We consider objectives in the safety fragment of computation tree logic (CTL) [1]. This allows us to express objectives like: $AG \neg collision$, which says that “on all paths (A) globally (G) no collision should occur”. We will add such an example in the paper.
>
> The probabilities used in the paper ($P^{max}$, $P^{min}$ for MDPs, P for Markov Chains) can be computed using standard algorithms for solving MDPs. Many tools allow to compute these probabilities efficiently (like PRISM, STORM, or TEMPEST) using various algorithms, e.g., using sound value iteration (SVI) [2]], a dynamic programming approach, or a translation into a linear program. Thus, the estimates are computed using exact probabilities, when sound algorithms are used to compute $P^{min}$ and $P^{max}$.
> The core of SVI is the iterative computation of the probabilities using the transition matrix of $M^i$. The complexity therefore scales polynomially with the number of states in the MDP. The powerful tool support in the community allows to compute these probabilities for MDPs with tens of millions of states efficiently [3]. We will point this out more clearly in the paper.
> [1] C. Baier and J. Katoen, “Principles of Model Checking”.
>
> [2] T. Quatmann et al. “Sound Value Iteration”
>
> [3] J. Katoen, “The Probabilistic Model Checking Landscape”
>
> ## W2:  Selection and sensitivity to the parameters.
> We thank the reviewer for the comment. While we think that the used parameters are quite natural, we agree that a discussion of the selection and the effects of the parameters will improve the paper.
>
> $\delta_\phi$: The probability of visiting an unsafe state should be below some threshold $\delta_\varphi$. This threshold is part of the safety specification.  For example, the $P (AG \neg collision) <= \delta_\varphi$. The higher the threshold, the fewer risks the agent is allowed to take. Extreme cases: if $\delta_\varphi = 1$, the framework classifies an action as unsafe if there is any risk that safety will be violated. If $\delta_\varphi = 0$, any behavior is safe.
>
> $\epsilon_\phi$: IMT stops if the difference between the optimistic and the pessimistic safety estimate is below this threshold for all states. Thus, the smaller this threshold is selected, the more test cases have to be executed until the algorithm terminates, but the more accurate is the estimated expected probability of the agent's policy to satisfy safety.
>
> $\delta_i$: When using clustering, we want to cluster highly ranked states. Thus, this threshold defines the states that will be clustered and potentially tested. The lower the value, the more states will be clustered and potentially tested. The chosen value for $\delta_i$ in the skiing experiment is $0.8$ (Line 297) since this value successfully excludes states that are far from any potential safety critical states. We apologize for not using the defined variable.
>
> ## W3: Continuous states and action spaces.
> We will add a paragraph to discuss this extension. Please also see our answer in the global rebuttal.

---

### Official Review · Reviewer_sLgt · 2024-07-13

**Soundness:** 4
**Presentation:** 3
**Contribution:** 3
**Rating:** 6
**Confidence:** 4

**Summary:**

This paper presents a framework called importance-driven model based testing for RL models.  ​ It uses a model-based approach to compute estimates of safety based on the MDP explored so far using the policy and ranks the importance of states based on the impact of decisions on safety.  Then it samples the policy on these state to further restrict the MDP. ​ The framework provides formal verification guarantees and partitions the state space into safe and unsafe regions. ​ The authors also propose a version to cluster similar states to further reduce the number of samples but at the cost of a relaxed guarantees. Experimental evaluations on different environments of various sizes demonstrate its effectiveness in reducing number of samples needed compared to random sampling both with and without the model. ​

**Strengths:**

- **Relevance**: Policy testing is an important area of research in reinforcement learning.
- **Novelty**: The paper introduces a novel algorithm that uses model-based approaches to reduce sample complexity while providing strong guarantees.
- **Soundness and clarity**: The experiments are thorough. The results clearly illustrate the system's effectiveness and are presented clearly with appropriate images and plots. Figure 2, in particular, best illustrates the effectiveness of this system.

**Weaknesses:**

Some of the weaknesses of this paper are:
- The approach relies on very strict assumptions on having a model for the environment with fixed set of states and actions, which could be very restricting for real world RL tasks.

- The introduction and abstract could better motivate and establish the problem. Currently, they only explain the algorithm without much context. For example, it is not clear how the safety estimates are computed until sections 2 and 3 or why computing them iteratively is easier than sampling the action on every state.

**Questions:**

- How can number of policy queries in Fig 3c be greater than the number of states in the environment?

- For runtimes, it will also be useful to break it down into model checking and policy sampling times for both IMT and MT approaches.

- Please add legends to Figure 3

**Limitations:**

The authors briefly talk about limitation about the requirements of the current approach, but they could definitely expand a bit more on these limitations early on the paper.

---

> ### Author Rebuttal · Authors · 2024-08-07
>
> We thank the reviewer for the detailed evaluation of our paper and for seeing the relevance of our proposed testing approach. We believe that our approach has the potential to significantly impact testing for DRL and hope that it will lead to additional interesting follow up research. In the following, we will address the mentioned weaknesses and answer the questions posed.
>
>
> ## Q1: How can the number of policy queries in Fig 3c be greater than the number of states in the environment?
> One of the significant advantages of our model-based testing approach is that we only need to sample the policy once per test case, as the model immediately provides information about the safety of the selected action (including induced safety risks in the following steps).
>
> This is not the case in random testing. Since random testing does not have a model in the background, it cannot measure the induced safety risks of a particular decision. Thus, we sample the policy and simulate the environment for multiple steps. This is the predominant approach in (software) testing, where a system is executed for a predetermined number of steps in order to reveal defects encountered along the way.
>
> Hence, for every test from a selected state, we query the policy several times.
> ## Q2: Breaking down the runtimes into times for model checking and policy sampling and Q3: Add legends to Figure 3
> For both Q2 and Q3 we thank the reviewer for the comment and will extend the paper to make the requested adjustments.

---

### Official Review · Reviewer_ak3E · 2024-07-13

**Soundness:** 2
**Presentation:** 3
**Contribution:** 2
**Rating:** 4
**Confidence:** 4

**Summary:**

This work looks into the RL testing problem. RL policies are complex and hard to understand in terms of safety and performance. This work aims to test the policies via states in which the agent’s decisions have the highest impact on the expected outcome. This paper proposes a model-based method to compute a ranking of state importance and focus the testing on the highest-ranked states. The evaluation covers multiple benchmarks and shows that this approach can find unsafe policies.

**Strengths:**

1. RL testing is important and few works touched on this.
2. The idea of using the highest impact states to test the policies is interesting and seems effective.

**Weaknesses:**

1. The guarantee about the correctness of the ranking does not seem to be clarified. Do you iterate over all the actions in the action set? What if the action space is large and continuous?
2. The testing is based on probabilistic model checking instead of worst-case checking. It would be good to bring out this in the abstract and introduction explicitly.

**Questions:**

1. How is the ranking computed? Do you cover continuous space?
2. What’s the scalability of this approach? Are there any cases where testing fails?

**Limitations:**

The limitations are not discussed in the main test.

---

> ### Author Rebuttal · Authors · 2024-08-07
>
> We appreciate the reviewer’s interest in our idea of guiding DRL testing through an importance ranking. We also agree that there is a current lack of testing methodologies for DRL. Additionally, considering the many potential extensions of our approach, we believe our method could significantly impact the development of DRL testing strategies. As it is the first paper that brings model based testing into DRL, we hope it will inspire follow up research.
>
> ## Question - Computation and correctness of the ranking (W1)
>
> The optimistic and pessimistic estimates are computed using the probabilities $P^{max}$ and $P^{min}$, which define the expected maximal and minimal probability to satisfy a property $\varphi$, respectively (see definition 3.1 paper). These probabilities can be computed with standard algorithms for solving MDPs. Many tools allow to compute these probabilities efficiently using various algorithms, e.g., using sound value iteration (SVI) [1], a dynamic programming approach, or a translation into a linear program. Thus, the estimates are computed using exact probabilities, when sound algorithms are used to compute the probabilities. Hence, if the MDP model is accurate, the provided estimates are exact.
>
> To compute the ranking, we indeed iterate over all the actions in the action set (we are currently not considering continuous action spaces. With our current approach, we would need to discretize the action space).
> The rank of a state s is given as the maximal difference between the optimistic estimates with respect to the available actions ($max_{a,a′} (e_{opt}(s, a, n) − e_{opt}(s, a′, n))$. $e_{opt}(s, a, n)$ is computed using the probabilities $P^{max}$.
>
> For a detailed discussion about the definition of the ranking, please see our answer to Q1 of  reviewer xBae.
>
> ## Question -  Scalability
>
> Being based on probabilistic model-checking, which internally uses value iteration,  the test-case generation scales to tens of millions of states in the MDP that serves as the test model [2].
>
> Please note that the size of the environment under test can be far larger. In model-based testing, a common strategy for large and complex systems is usually to create different models that capture different aspects of the system under test. Furthermore, to test for safety, it is often sufficient to consider abstract MDP model with a reduced feature space. States in an MDP typically encompass all relevant features of an agent and its environment. States then become vectors of feature values, where each combination of these constitutes a state. Not all features may be relevant to safety, though. By disregarding any features irrelevant to safety, the original model can be pruned to a
> much smaller model. Retaining the safety-relevant dynamics of the environment allows the use of model-based techniques for high-dimensional environments.
>
> Thus, it is often possible to construct MDP models with a manageable size (tens of millions of states are fine) that are suitable for safety testing, even if the environment under test is continuous or very high dimensional. This concept is also deployed in other areas like enforcing safe exploration during training via shielding [3]. As soon as such an MDP is available, our testing approach can be deployed.
>
>
> ## Question -  Continuous states and actions spaces
>
> We refer to our answer in the global rebuttal.
>
> ## Question - Are there any cases where testing fails?
>
> When using the approach in Section 3.1, our approach provides verification guarantees: Thus, after our testing framework terminates, the policy is verified to adhere to the safety requirements.
> As soon as clusters are used, the formal verification guarantees are lost, and unsafe behavior might be missed, even if the algorithm terminates. The advantage of using our testing approach is that similar states are clustered: Thus, states close to a certain safety hazard are clustered together and the agent's behavior with respect to this hazard is tested several times (even if not exhaustively). Unsafe behavior is only missed if all tested states within a cluster are found to be safe, but there is an untested state within the cluster from which the agent selects an unsafe action.
> Note that as soon as we detect a single unsafe behavior in a cluster, all states in the cluster are marked as unsafe.
>
> [1] Quatmann et al. “Sound Value Iteration”
>
> [2] Katoen. “The Probabilistic Model Checking Landscape”
>
> [3] Jansen et al. ”Safe Reinforcement Learning Using Probabilistic Shields”

---

### Official Review · Reviewer_nWP9 · 2024-07-14

**Soundness:** 3
**Presentation:** 3
**Contribution:** 2
**Rating:** 6
**Confidence:** 3

**Summary:**

The paper proposes an algorithm (IMT) to test a deterministic policy in finite MDPs where a model of the MDP is available. The primary contributions of the paper are algorithmic and empirical. IMT works by iteratively partitioning the state space into safe, unsafe or undetermined. Using the MDP model, IMT is able to construct optimistic and pessimistic bounds on the property being tested (e.g., safety). These estimates are used to identify the states with the largest gap in their optimistic estimates between any pair of actions. IMT prioritizes testing these states next, which has the effect of tightening the bounds and taking the largest possible step towards convergence and halting. The final result includes a partitioning of the state space into safe and unsafe states, which is valuable feedback for improving the policy (offline). The paper also includes a clustering-based variant to scale the algorithm to larger MDPs. Experiments on 3 domains demonstrate the utility of IMT against a random baseline and a variant which does not rank the states.

**Strengths:**

+ The paper studies the important problem of verifying black-box policies used in control applications. This is an important question as RL-learned controllers are deployed more widely. Advances here are likely to be of significant interest to the community.

+ The paper is well written. The main ideas, algorithms, ideas and experiments are generally explained clearly. Although some important implementation details are missing, the paper is nevertheless easy to read.

+ The proposed algorithm is intuitively clear and using the estimates to identify "important" states to test next seems to work well. In cases where a model of the discrete MDP is available (or perfectly learnable), this seems like a useful technique to rigorously test a policy and uncover states where the policy does not behave safely. The experiments and appendices include a good amount of detail.

**Weaknesses:**

- The paper does not provide sufficient detail on important algorithmic implementation details. Examples include the prerequisites for the algorithm (simulators, CTL), implementation of `computeEstimates`, $e_opt(s, a, n)$, clustering similarity/distance function, etc.

- The assumption of a correct model of the MDP is a somewhat large one in many domains. A discussion of the effort needed to obtain such a model would improve the paper and make its applicability to new domains clearer. Additionally, exploring the use of noisy simulators with varying levels of noise in the dynamics would be really interesting to see. As the paper mentions in Line 331, being able to leverage models of the environment learned from data is likely to increase the applicability. However, these are likely to include noisy dynamics so an experimental evaluation of the impact of noisy dynamics in the current experimental setup would have strengthened the paper.

- A comparison to stronger baselines would make it easier to assess the empirical contributions of the paper. Currently, the only comparison is to random testing. Comparing safety-violation detection rates against black-box policy testers (e.g., MDPFuzz, STARLA) using the same budget would have helped place this paper in the larger body of work on DRL testing as well as understand the applicability of these techniques in domains with different characteristics (size, model availability or learnability, etc.).

**Questions:**

- What is the exact implementation of `computeEstimates`? How does it scale with the size of the input (restricted) MDP? Similarly, how exactly is $e_opt(s, a, n)$ estimated?

- What steps are involved in adding a new domain from scratch? How easy or hard is it to create the simulator, specify the CTL objective, etc.?

- Is it feasible to quantify or conduct investigations into how AMT might handle noise in the transition function in the current experimental setup?

- What is the precise definition of a robust policy in Line 195?

- What is the computational cost of the clustering approach used here? Should it be reflected in the complexity analysis in Lines 214 - 219?

- How does the quality of the resulting clusters affect performance? Does Alg 2 degrade gracefully as cluster quality decreases?

- Is it necessary to use a different mechanism (sink states) to restrict the MDP when clustering? How does terminating the trajectory at the sink state affect the e_opt, e_pes estimates, if at all? Does the approach still work if the restriction mechanism from Alg 1 is used?

- Does it make sense to compare safety-violation detection rates against other baselines (MDPFuzz, STARLA)?

- Does it make sense to consider continuous state and / or action MDPs using this framework? If yes, what modifications might be needed to make it work on cMDPs?

**Limitations:**

Yes

---

> ### Author Rebuttal · Authors · 2024-08-07
>
> We thank the reviewer for the detailed evaluation of our paper. We believe that our proposed testing framework has the potential to significantly impact testing for DRL. Although testing DRL is intrinsically challenging, the problem has only recently garnered attention from the testing community. Currently, there is no common framework for RL testing methods, and most papers explore search-based strategies. Our paper is the first to introduce model-based testing into the DRL setting, a method that has proven highly relevant in many other domains and we believe has much potential also in the DRL setting.
>
> ## Q1: Details on computation of estimates, scalability, and what is estimated.
>
> The optimistic and pessimistic estimates are computed using the probabilities $ P^{max} $ and $P^{min}$, which define the expected maximal and minimal probability to satisfy a property $\varphi$, respectively (see definition 3.1 paper). These probabilities can be computed with standard algorithms for solving MDPs. Many tools allow to compute these probabilities efficiently using various algorithms, e.g., using sound value iteration (SVI) [1], a dynamic programming approach, or a translation into a linear program. Thus, the estimates are computed using exact probabilities, when sound algorithms are used.
> The core of SVI is the iterative computation of the probabilities using the transition matrix of $M^i$. The complexity therefore scales polynomially with the number of states. The powerful tool support in the community allows to compute these probabilities for MDPs with tens of millions of states efficiently [2].
>
> [1] Quatmann et al. “Sound Value Iteration”
>
> [2] Katoen. “The Probabilistic Model Checking Landscape”
>
> ## Q2: Steps for adding a new domain.
>
> Our framework requires an MDP model of the environment and a CTL specification. The work involved in adding a new domain mostly amounts to creating an MDP. That is, you need knowledge about the environment dynamics on an abstract level. Most probabilistic model checkers accept inputs expressed in the symbolic modeling language PRISM [3]. For simulation, we generally rely on a gymnasium-like interface with a step and a reset functionality.
> The objectives specified in CTL are relatively easy to define. For safety, it is generally sufficient to define events that must not happen. We also refer to our global rebuttal.
>
> [3] Kwiatkowska. “PRISM 4.0: Verification of Probabilistic Real-time Systems.”
>
> ## Q3: Consequences of noise in transition relation.
>
>
> IMT is only little affected by noise in the transition probabilities: the noise might only change the order in which states are tested. Let’s assume that from a certain state, a transition wrongly underrepresents the probability of traversing to an unsafe state. Our testing framework might then assign a lower rank to this state. However, if our algorithm is executed until convergence, the state will eventually be tested.
> Our approach benefits from being a testing framework, with its primary task being the automatic selection of test cases. Even if the MDP model used to select the next test cases is not perfect, our approach will likely still identify interesting test cases. We thank the reviewer for this question and will emphasize the robustness of our approach against noise in the paper.
> ## Q4: Definition of a robust policy in Line 195.
> We consider a policy to be robust if small changes in the input do not significantly change the chosen action. We mention this when discussing the testing approach with clustering: The intuition is that if we assume that the states in the cluster are similar and the policy makes similar decisions in similar states, testing only a fraction of the states within each cluster yields good testing results.
> ## Q5: Costs for clustering.
> For the complexity analysis, we focused on the steps of the approach proposed in the paper. We refrained from including clustering because it is problem-dependent. However, as we applied k-means, which scales very well, we found that its runtime is negligible in practice.
> ## Q6: Impact of the clusters' quality on performance
> Low-quality clusters can lead to an unnecessarily large testing effort or missed unsafe behavior.
>
> Consider the case when a cluster is identified as unsafe: If the cluster is too large and includes states that are actually safe, our iterative testing framework will needlessly cluster and test the predecessor states of the cluster. For example, in the context of skiing, our framework would end up testing states that are not critical for avoiding collisions with trees or poles. Now, consider a scenario where all states in a cluster are deemed safe: If the clustering method groups states that are not sufficiently similar, our testing approach may fail to detect unsafe behavior in the policy.
> Please note that we provide verification results (guarantees that any unsafe behavior will be detected) only when no clustering is used.
> ## Q7: Necessity and impact of sink states.
> Yes, it is necessary to adapt the restriction of the MDP in Alg2.
> If a single state of a cluster is assigned a failing verdict, we mark all states of this cluster as failing states and turn them into sink states. This allows the generalization of testing results to whole clusters, increasing scalability since not all states of a cluster need to be queried.
> Terminating the trajectories at the sink states in clusters generally decreases the values of both estimates. By following a conservative approach, we consider all states in a cluster as unsafe if even one state is assigned a failing verdict. As a result, the estimates of the MDP with sink states are generally smaller than they would be in the unrestricted MDP. We will add a discussion to the paper.
> To use the approach from Alg1, test executions for all states in a cluster would be necessary, which contradicts the goal of minimizing the testing budget.
> ## Q8 and Q9
> We refer to our answer in the global rebuttal.

---

> > ### Comment · Reviewer_nWP9 · 2024-08-13
> > **Re. author response**
> >
> > I thank the authors for their detailed response to the reviewers. After reading the response, other reviews and comments, I've increased my score to "Weak Accept". While I don't think there's anything technically wrong with the paper, there are a number of major assumptions being made, which might reduce the applicability of the proposed approach. The performance of the proposed method is not explored on abstract or learned models of the environment, which is the most likely scenario. Including these experiments would significantly strengthen the paper, in my opinion. In the absence of these, it becomes challenging to evaluate the paper for impact and overall utility.

---

### Author Rebuttal · Authors · 2024-08-07

We sincerely thank the reviewers for their valuable feedback. Our paper introduces model-based testing to the reinforcement learning setting.  Since this is the very first paper in that direction, we agree with the reviewers that there are numerous intriguing directions yet to be explored, such as continuous state and action spaces. We hope that this work will inspire future research in model-based testing for deep reinforcement learning (DRL).

In this global rebuttal, we aim to address the questions and concerns raised by multiple reviewers.

## The approach relies on the assumption of having an MDP model of the environment.

We agree that needing an MDP model of the environment may be considered a hurdle toward the adoption of model-based testing (MBT). Indeed, it is often perceived as such in industrial applications of MBT for testing of conventional software. However, experience reports [1,2] from industry reveal benefits resulting from the activity of modeling software systems and their environments beyond just enabling test-case generation.  The benefits include a better understanding of the system, environment, and requirements. The model commonly resolves ambiguities and helps to communicate between different stakeholders. It also opens venues for other activities such as the verification of temporal properties. In RL, environmental models enable safe exploration during training via techniques like shielding. Furthermore, the application areas of neurosymbolic AI are constantly growing, utilizing the same concept of combining model-based symbolic AI (e.g., to guarantee safety) with subsymbolic AI (e.g., to achieve high scalability and performance). Additionally, the software engineering community showed several successful applications of model learning to create test models automatically, thus circumventing manual modeling.
This paper is the first to introduce MBT into the testing of reinforcement learning. In line with related model-based techniques in other areas, we believe the benefits—such as using the model to select the most important test cases, reducing the number of policy samples, and computing estimates over the entire state space—outweigh the drawback of needing a model.



[1] Emil Alegroth et al. “Practitioners’ best practices to Adopt, Use or Abandon Model-based Testing with Graphical models for Software-intensive Systems”, Empirical Software Engineering (2022) 27: 103

[2] Robert V. Binder et al. “Model-Based Testing: Where Does It Stand?”, COMMUNICATIONS OF THE ACM (2015), 58: 2

## Extension to continuous state and action spaces

Indeed, extending our framework to environments with continuous state and action spaces is possible, and we plan to pursue this as our next step. We plan to learn abstract finite-state models of the environment. To automatically learn such a model, we are currently evaluating two potential approaches: (1) Compute an abstract state-space of the MDP model representation through dimensionality reduction and clustering of observed environmental states. The stochastic transitions are learned via a combination of active and passive model learning, as proposed in [5].
(2) Apply world model learning via neural networks, as in DreamerV3 [4], from which we extract finite-state models via quantized bottleneck insertion as in [3].
In this paper, we provide the first step: We introduce the notion of importance-driven testing and show its potential via the Atari Skiing game. To handle continuous domains, we will apply importance rankings in abstract environment models. We believe that the learned MDP models will be sufficiently precise to compute good importance rankings across all states, thereby identifying critical situations that need to be tested. Thus, we believe our method will evolve into a valuable testing method for intricate control policies in complex environments.

[3] Anurag Koul et al. "Learning Finite State Representations of Recurrent Policy Networks"

[4] Danijar Hafner et al. "Mastering Diverse Domains through World Models"

[5] Martin Tappler et al. "Learning Environment Models with Continuous Stochastic Dynamics"

## Comparison to stronger baseline

A comparison to existing search-based testing is difficult for two reasons. (1) There is no common framework for RL testing methods yet. For example, MDPFuzz only fuzzes the initial states, while STARLA creates complete episodes that must end in a terminal state. Therefore, our approach would be hardly comparable due to the difference in setting. (2) A fair comparison would likely need to go beyond safety-violation detection rates, which has been the focus of existing work, toward more comprehensive criteria. For example, STARLA might create different test cases for the same failure states, i.e., different episodes that traverse a common subset of states. As a result, STARLA might have an artificially high failure-detection rate, although certain states are covered multiple times, which we strictly avoid. In our comparison to random testing, we check how many of the failure states are found, which is not easily enforceable in the meta-heuristic search performed by STARLA.
Hence, we need suitable quality metrics for test cases in RL, which we are currently exploring in addition to model learning. Note that quality metrics (called adequacy criteria in software testing) and test-case selection criteria are two sides of the same coin, thus our approach presents a step toward defining proper quality criteria.

---

### Decision · Program_Chairs · 2024-09-25

**Decision:**

Accept (poster)

**Comment:**

The paper introduces a novel technique for safe RL leveraging differentiable NN verification tools. The algorithm uses the $\alpha,\beta$-crown verification method.

The reviewers agreed that the approach is novel, and the authors addressed their main concerns during the rebuttal phase. The main concern was the assumption of having an environment model at hand. Nonetheless, it seems to be the first successful application of Model-Based Testing in RL; there are numerous applications where the world model is accessible.

I therefore accept the paper for the NeurIPS conference.

In the final version of the paper, please include the discussion on the neural network-controlled systems, the missing related work, and the additional benchmarks and clarifications provided in the response to the reviewers.